# Enhancing smart charging in electric vehicles by addressing paused and delayed charging problems

Nico Brinkel [1] ✉, Thijs van Wijk[2], Anoeska Buijze [3], Nanda Kishor Panda [4], Jelle Meersmans[5], Peter Markotić[2], Bart van der Ree[6], Henk Fidder[7], Baerte de Brey[2,7], Simon Tindemans [4], Tarek AlSkaif [8] & Wilfried van Sark [1]

Smart charging of electric vehicles can alleviate grid congestion and reduce charging costs. However, various electric vehicle models currently lack the technical capabilities to effectively implement smart charging since they cannot handle charging pauses or delays. These models enter sleep mode when charging is interrupted, preventing resumption afterwards. To avoid this, they should be continuously charged with their minimum charging power, even when a charging pause would be desirable, for instance with high electricity prices. This research examines this problem to inform various stakeholders, including policymakers and manufacturers, and stimulates the adoption of proactive measures that address this problem. Here, we demonstrate through technical charging tests that around one-third of tested car models suffer from this issue. Through model simulations we indicate that eliminating paused and delayed charging problems would double the smart charging potential for all applications. Lastly, we propose concrete legal and practical solutions to eliminate these problems.

With the growing adoption of Electric Vehicles (EVs), our transportation and electricity systems are becoming increasingly intertwined. Initially, a network of petrol stations provided the energy requirements to fulfill our road transportation needs. However, the electricity grid infrastructure increasingly takes over this role through EV charging[1,2]. This transition brings new challenges to the electricity system, particularly to the grid infrastructure. Most EV charging occurs in Low-Voltage (LV) grids at home or on-street charging stations[3], and the majority of these grids were designed decades ago without the concept of EV charging in mind. The charging power of an EV is significantly higher than the typical peak-time power consumption of a household, and since most EV users tend to arrive at their charging station at a similar time,

concentrated charging moments are expected in residential LV grids[4,5]. As a result, EV charging is likely to cause grid congestion[6–8].

Grid reinforcements could serve as a solution, but their feasibility is hindered by the exorbitant costs[9,10] and a shortage of qualified personnel to execute these reinforcements[11,12]. Another approach to alleviate grid congestion is to move away from uncontrolled charging, where EVs charge at maximum power upon arrival until their demand is met. For most EV charging sessions, the connection time to a charging station considerably exceeds the required time to meet their charging demand. This provides ample opportunities for EV smart charging. With smart charging, EV charging sessions are optimized for different objectives by aligning the charging moments and charging

[1]Copernicus Institute of Sustainable Development, Utrecht University, Princetonlaan 8a, 3584 CB Utrecht, The Netherlands. [2]ElaadNL, Westervoortsedijk 73, 6827 AV Arnhem, The Netherlands. [3]Faculty of Law, Economics and Governance, Utrecht Centre for Water, Oceans and Sustainability Law, Utrecht University, Newtonlaan 201, 3584 BH Utrecht, The Netherlands. [4]Department of Electrical Sustainable Energy, Delft University of Technology, Mekelweg 4, 2628 CD Delft, The Netherlands. [5]Enervalis, Lummense Kiezel 51, 3500 Hasselt, Belgium. [6]Utrecht Sustainability Institute, Postbus 85057, 3508 AB Utrecht, The Netherlands. [7]Stedin Groep, Blaak 8, 3011 TA Rotterdam, The Netherlands. [8]Information Technology Group (INF), Wageningen University and Research (WUR), 6706 KN Wageningen, The Netherlands. ✉e-mail: n.b.g.brinkel@uu.nl

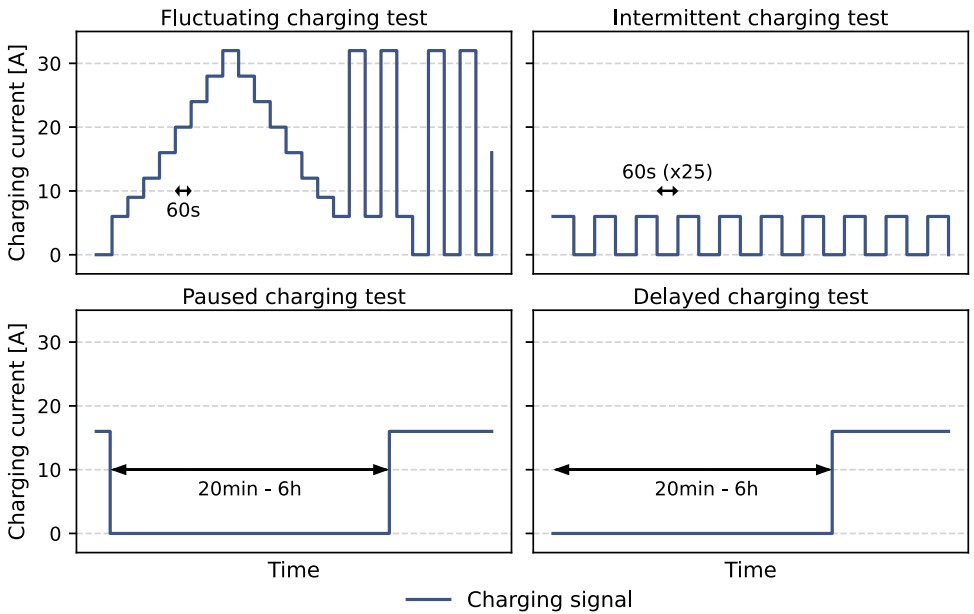

**Fig. 1 | Overview of the different smart charging tests in the charging protocol at the ElaadNL Testlab.** The plots provide insight into the duration and amperage of the charging signals that are sent to the EV during the different smart charging tests that are considered.

speed over time with user preferences and current market or grid conditions[3,13].

EV smart charging can benefit both grid operators and EV users and can facilitate the ongoing energy transition. Different studies showed that the application of smart charging could support grid operators in mitigating grid congestion and power quality problems (e.g., refs. [14–17]). Similarly, smart charging can be applied for the provision of balancing reserves to Transmission System Operators (TSOs)[18,19]. It can also help EV users reduce their charging costs by taking advantage of moments with low electricity market prices when considering static or dynamic Time-of-Use (ToU) pricing schemes[20,21]. Lastly, the roll-out of smart charging can accelerate the energy transition by shifting the charging demand of EVs to moments with excess renewable generation[22–24], thereby reducing the dependency on fossil-based energy resources and mitigating the intermittency challenges associated with renewable energy sources. If vehicle-to-grid (V2G) functions are considered, EVs could even act as a mobile storage medium for excess renewable energy, which can be utilized to meet the electricity demand during periods of renewable energy shortage.

However, it is not widely known that the current deployment of smart charging is hindered by the technical capabilities of EVs. As this research will show, a significant share of EV models (both Plug-in Hybrid EVs (PHEVs) and Battery EVs (BEVs)) is unable to perform paused or delayed charging. In paused charging, the charging process is interrupted after the EV was previously charging, while in delayed charging, the start of the charging process is postponed after the EV arrives at the charging station. Both processes cause a substantial portion of the EV models in the market to switch to sleep mode. This makes them unresponsive to charging signals after the pause or delay, posing a risk of unmet charging demand at their departure from the charging station. To avoid this, EVs need to be continuously charged with at least their minimum charging current[25], even when this is not desirable, for instance at moments with a high electricity price or grid load. Consequently, the paused and delayed charging problems of EVs reduce the potential impact of smart charging.

Remarkably, the technical problems associated with EV smart charging have hardly been addressed in scientific literature and the media, leading to low awareness about these issues among different stakeholders, including policymakers, EV manufacturers and grid operators. This is evident from several factors. First, as this research

will show, a notable portion of newly-introduced EV models cannot perform paused or delayed charging, indicating that EV manufacturers may not be aware of this issue. Second, almost all smart charging studies fail to consider that a considerable share of the EV fleet cannot perform paused or delayed charging, resulting in an overestimation of the smart charging potential. Third, no legal or policy initiatives appear to address this problem.

In this work, we shed light on these problems to raise awareness among relevant stakeholders (e.g., grid operators, EV manufacturers and policymakers) about the prevalence of technical smart charging problems and their impact on the effectiveness of smart charging. We present the results of large-scale technical charging tests, which indicate that around one-third of the EV models in the market cannot handle charging pauses or delays. Moreover, this study presents the results of model simulations that quantify the impact of EVs' inability to perform paused or delayed charging on three different applications for which smart charging can be used, namely: i) charging cost reduction, ii) mitigation of grid congestion, and iii) offering flexibility products to grid operators. The outcomes of these model simulations show that the potential impact of smart charging is halved for all applications if paused or delayed charging cannot be considered. Lastly, the current international regulations and standards on this topic are discussed, and options to eliminate paused and delayed charging problems are analyzed.

## Results
### Technical smart charging tests
The technical performance of new EV and charging station models is evaluated using charging tests at the Testlab of ElaadNL, a knowledge center on EV charging established by Dutch grid operators. Manufacturers of EVs and charging stations are invited to test the technical performance of their products. They are tested on their interoperability, impact on power quality and ability to perform smart charging[26] using a standardized testing protocol to ensure comparability of results between charging tests for different EV models. The charging tests have been performed on a large share of the PHEV and BEV models on the Dutch market and manufacturers can use the results of these tests to improve the technical performance of their products. This section focuses on the charging test results related to different smart charging applications. It is important to note that some

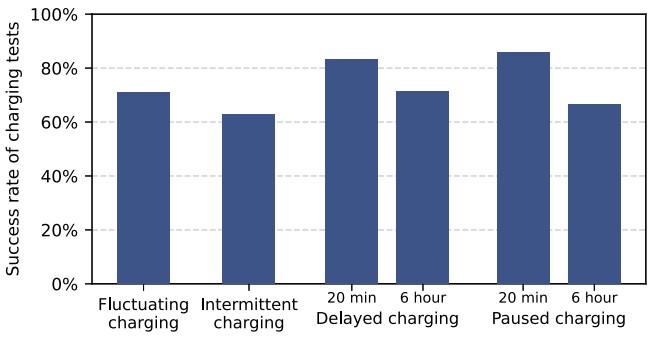

**Fig. 2 | Results of the conducted charging tests at the ElaadNL Testlab between 1 June 2020 and 1 January 2023.** Results are presented for fluctuating charging tests on 52 EV models, intermittent charging tests on 43 EV models, 20-min paused and delayed charging tests on 42 models and 6-h delayed and paused charging tests on 21 models. Source data are provided as a Source Data file.

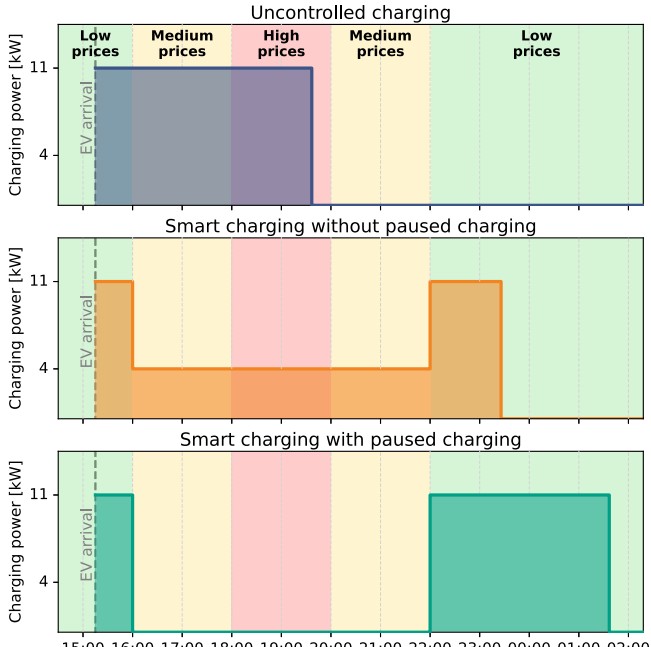

**Fig. 3 | A comparison of charging schedules of one charging session with a simplified cost-optimization approach for different charging regimes.** Charging session characteristics are an arrival time of 15:15 (dashed line in Figure), a departure time of 08:00, a charging demand of 48 kWh and a minimum and maximum charging power of 4 kW and 11 kW, respectively.

manufacturers have rectified the charging issues after undergoing the charging tests through the installation of software updates.

The smart charging tests conducted at the Testlab aim to evaluate the EV's response when exposed to various charging profiles that could occur with different applications of smart charging. The parameters of the charging tests have been determined in consultation with different stakeholders, such as grid operators and EV manufacturers. The fluctuating charging test assesses whether an EV can handle smart charging applications with high fluctuations in the charging signal, such as solar charging (i.e., directly linking the charging power to the solar generation of photovoltaic (PV) systems). As shown in Fig. 1, this test considers a fluctuating charging signal between 6 and 32 amperes at a 60-s interval. These current values correspond to the prescribed minimum and maximum charging currents for EV charging, as defined in the International Electrotechnical Commission's communication standard for EV charging[27]. The intermittent charging test evaluates the EV's response to a charging session with a high number of charging pauses, which is for instance relevant when applying smart charging for load balancing at car parks (i.e., quickly alternating the charging power between charging stations to reduce the peak charging power of the car park). In this test, the charging signal is switched 25 times between 0 and 6 amperes at 60-s intervals. The last set of tests analyses the EV's response to paused and delayed charging. These tests are relevant for smart charging applications that require longer periods without charging, such as smart charging to reduce charging costs with static or dynamic ToU tariffs or smart charging to mitigate grid congestion. The paused and delayed charging tests have a similar setup. The paused charging tests assess the EV's ability to properly react to the charging signal after a charging pause, which is implemented after the vehicle has been charged for a brief period. The delayed charging tests also consider a charging pause, which starts directly after the EV arrives at the charging station. Both tests are conducted with pauses of 20 min and 6 h.

Figure 2 presents the results of smart charging tests that were conducted with 52 EV models (cars, vans and motorcycles) between 1 June 2020 and 1 January 2023. A charging test was labeled as unsuccessful if the tested EV model ceased charging or if its charging current violated the current limits specified in the EV charging standards by exceeding the charging signal by at least 0.5 amperes[27]. The latter problem only occurred with the fluctuating and intermittent charging tests. The success rate of the fluctuating charging test equals 71% for the tested EV models. The share of tested models that can follow the intermittent charging profile is lower and equals 63%. In both cases, charging problems were observed in both PHEVs and BEVs, with the majority of failed tests attributed to violations of current limits. These results indicate that a large share of the tested EV models is unable to perform smart charging for applications at which the charging power could fluctuate rapidly. These issues appear to be caused by the software settings of

certain EV models, which identify charging stations with high fluctuations in the charging signal as faulty without considering that the application of smart charging may cause these fluctuations.

The share of EV models that can deal with paused and delayed charging profiles depends on the pause duration. When considering a charging pause of 20 min, the success rate for the tested EV models equals 86% and 83% for paused and delayed charging, respectively. When the pause duration is extended to 6 h, the share of tested EV models that successfully pass the charging test reduces to 71% and 67% for paused and delayed charging, respectively. These problems manifested with both PHEVs and BEVs. These issues are also software-based: to prevent the 12-volt battery that powers the vehicle's electrical systems from draining, the EVs switch to sleep mode if no charging signal is received for an extended duration.

## Impact on smart charging's charging cost reduction potential

The inability of EVs to perform paused or delayed charging can diminish the effectiveness of different smart charging applications, including smart charging for participating in an electricity market that considers static or dynamic ToU tariffs (e.g., day-ahead electricity market)[28–30]. Charging costs can be reduced by shifting the charging demand from moments with high prices to moments with low prices. While longer charging pauses may be desirable at specific moments, for instance at moments with high electricity prices, they cannot be implemented into EV charging schedules as some EV models will shift into sleep mode and will become unresponsive to charging signals. Since the EV model is not specified in the current communication protocols between the EV, the charging station and the back-office of the charge point operator[27,31], charging pauses are generally not considered for all EVs, regardless of whether they are able to perform paused and delayed charging or not. The only way to reduce the impact of EV charging at the moments at which this is desired is by charging with the lowest possible current of the charging system, which is typically 6 amperes[27]. Figure 3 illustrates how this reduces the effectiveness of smart charging. It reports the charging schedules for

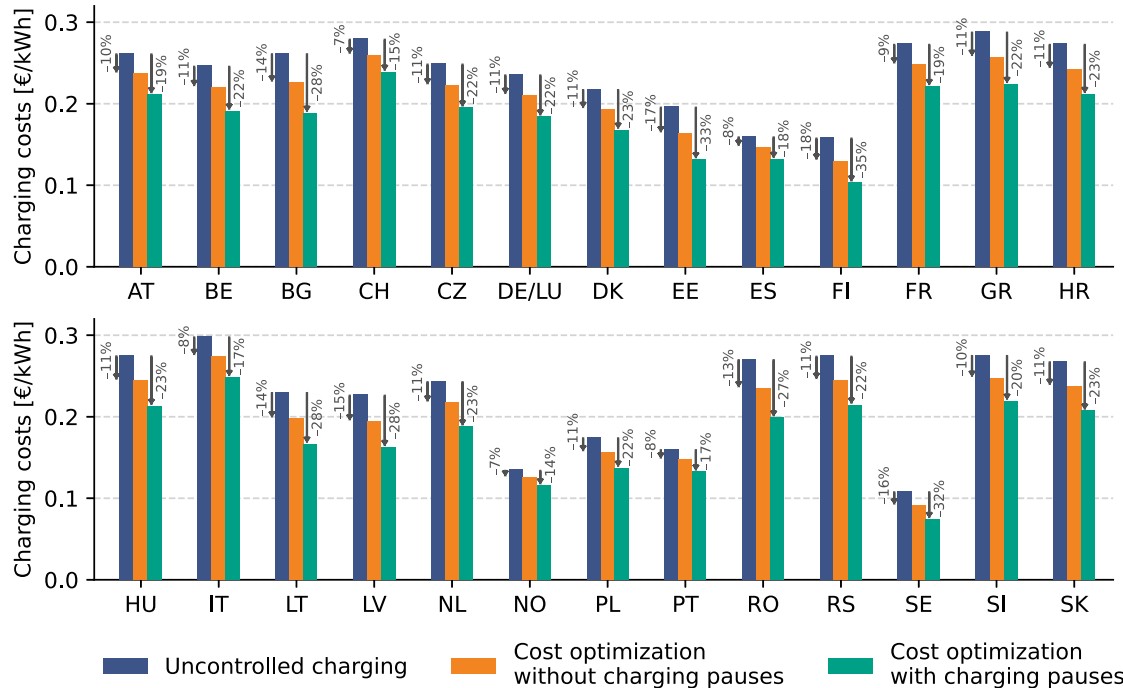

**Fig. 4 | Average EV charging costs for three different charging strategies in the day-ahead market of different European countries.** Results are presented for 322 public charging stations in residential areas for an assessment timeframe of one year between 1 February 2022 and 1 February 2023. Percentage values represent the cost decrease compared to uncontrolled charging (blue bars). For countries with multiple bidding zones (DK, NO, SE & IT), the average charging costs for all bidding zones are reported. As Germany and Luxembourg comprise one bidding zone, the results for these countries are reported together. Source data are provided as a Source Data file.

one charging session when considering different charging regimes (uncontrolled charging, smart charging without paused charging and smart charging with paused charging) for a simplified cost-optimization problem. This figure shows that if smart charging is applied without considering charging pauses, still a considerable share of the charging demand (50% for the charging session in Fig. 3) has to be fulfilled at moments with medium or high prices. This is in contrast to smart charging with paused charging, where the charging demand can be completely fulfilled at moments with low prices in this example.

Model simulations are performed to quantify the impact of the paused and delayed charging problems on the cost-reduction potential when using smart charging for electricity market bidding. Figure 4 presents the charging costs for a large EV fleet when participating in the day-ahead market in different European countries with perfect foresight. Three charging scenarios are considered in the model simulations: i) uncontrolled charging, ii) cost-optimization without considering paused or delayed charging and iii) cost-optimization considering delayed or paused charging. The results in Fig. 4 indicate that the inability to perform paused or delayed charging almost halves the cost-reduction potential of smart charging. When charging pauses can be considered, the cost-reduction potential compared to uncontrolled charging equals 14–35%, depending on the country. For all considered countries, the cost reduction potential is approximately half as high when no paused or delayed charging can be considered. This is because EVs are forced to charge with the minimum current of 6 amperes at times of high prices and cannot benefit from lower prices since their charging demand is fulfilled before those times arrive.

**Impact on smart charging's grid congestion control potential**

Smart charging can also be used to address grid congestion problems induced by EV charging by shifting the charging from moments with high local grid load to moments with low local grid load[32,33]. When charging pauses cannot be considered, EV charging cannot be completely shifted away from peak hours. Consequently, grid congestion problems will manifest at lower EV adoption levels when deploying

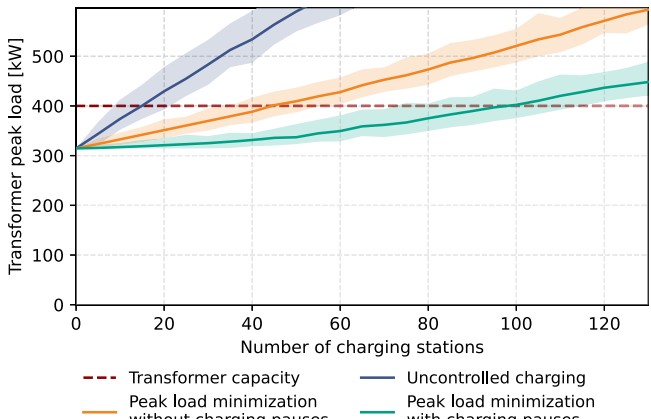

**Fig. 5 | Peak transformer load values in a one-year assessment timeframe when applying a peak transformer load minimization algorithm to a different number of installed on-street charging stations in one residential LV grid in the city of Utrecht, the Netherlands.** The analysis is repeated 100 times for each considered number of EV charging stations, with a randomly sampled subset of EV charging stations in each run. The line shows the average outcome for all model runs and the shaded area shows the 95% confidence interval. Source data are provided as a Source Data file.

smart charging without paused or delayed charging compared to the deployment of smart charging with these features.

Model simulations were conducted using a transformer peak load minimization algorithm for EV charging to investigate the impact of the paused and delayed charging problems on the potential for mitigating grid congestion through smart charging. Simulations were performed for both transformer peak load minimization with and without considering paused and delayed charging.

Figure 5 presents the transformer peak load values (i.e., the sum of the non-EV load and EV load) for a varying number of on-street

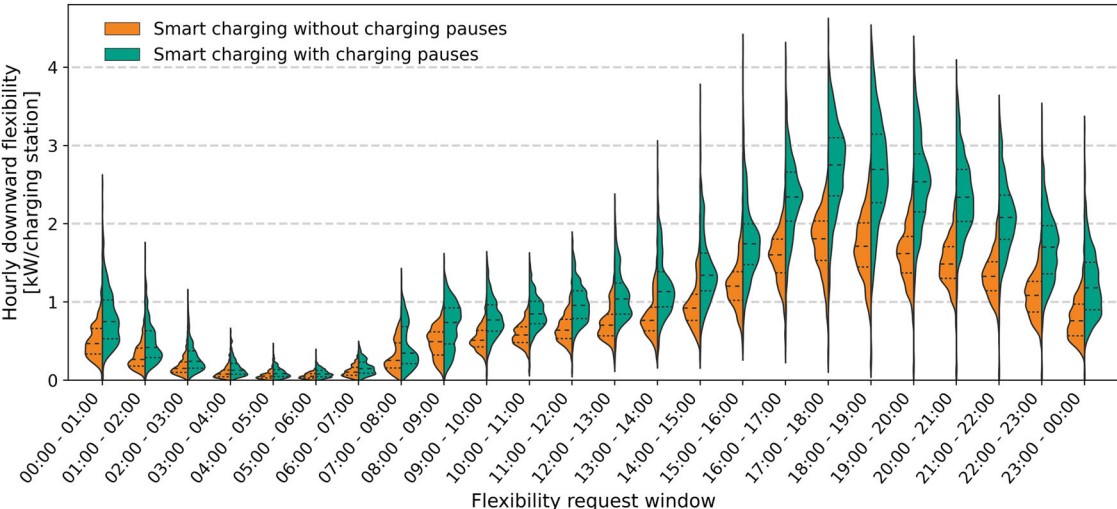

**Fig. 6 | Violin plot with the distribution of the available downward flexibility of an EV fleet for each hour of the day during the assessment timeframe of one year when considering the uncontrolled charging profile as the reference charging profile.** Results are compared for the case with and without paused and delayed charging. Model simulations are performed using historical charging data from 322 on-street charging stations located in the city of Utrecht, the Netherlands. The dashed and dotted lines in the violins represent the 25%, 50% and 75% quantile values. Source data are provided as a Source Data file.

charging stations connected to one LV grid. From this figure, one can determine the maximum number of EV charging stations that can be theoretically hosted in one LV grid without causing transformer congestion. The results show that the number of charging stations that can be installed in the considered grid without exceeding the transformer capacity is approximately twice as high when charging pauses can be implemented in the charging schedules. Transformer congestion problems will occur in the studied grid if it hosts between 35 and 55 charging stations, a peak load minimization algorithm is applied and charging pauses and delays cannot be implemented. This increases to a range of 75 to 115 charging stations when paused and delayed charging can be considered.

### Impact on smart charging's flexibility services potential

Moreover, smart charging can be applied to offer flexibility services to grid operators, for instance by supplying balancing reserves (e.g., frequency restoration reserves) to TSOs for restoring the balance between supply and demand[34–37]. Additionally, DSOs have been experimenting with local flexibility markets to address grid congestion issues[38]. The paused and delayed charging problems affect the amount of downward flexibility (i.e., a reduction in charging power from the reference charging schedule) that can be provided using smart charging. This was demonstrated through the final series of model simulations, which compared the available downward flexibility of an EV fleet under two scenarios: one that does not consider charging pauses and another that does. In these simulations, the maximum reduction in charging power from the reference charging schedule (uncontrolled charging in this case) is determined while ensuring that the charging demand of every session is met at departure.

The violin plot in Fig. 6 presents the distribution of the hourly available downward flexibility throughout one year for both considered cases. It shows that the available downward flexibility is higher during early evening hours. This is because more EVs are typically charging during this time in residential areas, resulting in a higher potential for charging power reduction. As visible in Fig. 6, the inability to perform charging pauses reduces the available downward flexibility, since EVs that cannot handle charging pauses cannot fully ramp down their charging power to provide downward flexibility and must keep the charging current above the minimum charging current. On average, 34% less downward flexibility can be offered when no charging pauses can be considered (95% CI: 24–44%).

### Options for eliminating charging problems

The previous sections emphasized the necessity of EV car models to be able to perform paused and delayed charging. This section examines existing international regulations and standards on this issue and explores solutions to avoid such problems in the future.

Two standards for EV charging currently address the delayed and paused charging problems. However, EV manufacturers are not obliged to comply with them. IEC 61851[27] is a set of standards that contains use cases for EVs and charging stations on how to wake up EVs that shifted to sleep mode after a charging pause. This standard also contains safety standards, and manufacturers of EVs and charging stations that comply with it are deemed to comply with the Low Voltage Directive (LVD)[39]. This directive applies to all electrical equipment traded within the EU and aims to safeguard the health and safety of persons, animals and property (art. 1 LVD). It is based on self-assessment, and there is no notified body that intervenes with the conformity assessment procedure. For EVs and charging stations, compliance with IEC 61851 leads to the presumption of compliance with the LVD (art. 12 LVD). By complying with this standard, EVs are able to perform paused and delayed charging. However, EV manufacturers do not need to use this standard to comply with the LVD; as long as the safety standards in the LVD are met, the product can be traded on the EU market. Since the ability to handle charging pauses is of limited relevance for product safety, EVs are able to comply with the LVD without complying with IEC 61851.

Secondly, the paused and delayed charging problem is addressed in a newly-developed standard for communication between EVs and their charging station. The ISO 15118-20 standard[40] has been developed to enable bidirectional EV charging through V2G technology and has been implemented in a small number of EV models. It includes a use case to re-establish communication with an EV in sleep mode by sending a wake-up trigger. While the implementation of this standard would likely eliminate the paused and delayed charging problems for compliant EVs, there is no legal requirement for manufacturers to incorporate it in their models. As a result, the paused and delayed charging problems might persist for some EV models since manufacturers might not be incentivized to implement the standard in models that are unable to perform V2G functions.

If the public sector, including governments and regulatory agencies, deem that the paused and delayed charging problems are too severe to be resolved without intervention, there are multiple

enforcement or stimulation methods available. First, these organizations could stimulate manufacturers to comply with the existing standards that address this issue. This can be done by taking an active role in informing manufacturers about the importance of EVs being able to handle paused and delayed EV charging. Alternatively, the public sector has the option to establish an EV model certification program, where EV models that successfully completed a set of EV smart charging tests are granted a certificate, which could make the specific EV model more appealing to consumers.

The public sector can also enforce the elimination of paused and delayed charging problems by implementing regulations on this topic. This could be modeled after the type-approval system that is currently in place. Before getting access to public roads in the EU, all car models need to acquire type-approval[41], issued by a national approval authority. The type-approval tests assess whether the car models meet EU safety rules (e.g., crash tests) and noise and emission limits. Expanding these tests to include an evaluation of the technical charging capabilities of EVs would ensure that only EV models meeting technical charging standards are permitted on the road. If the public sector prefers not to incorporate technical charging tests into the type-approval process, they could introduce legislation that makes compliance with standards that address the paused and delayed charging problems, such as ISO 15118-20, compulsory, either through self-assessment or by requiring testing and approval by a national approval authority. Ideally, all discussed enforcement or stimulation methods should be implemented at an international level, within entities like the EU, to enhance efficiency and maintain consistency in policies across different nations.

As long as the paused and delayed charging issues are not resolved, smart charging operators can use workaround solutions to identify whether a specific EV can handle charging pauses. A pause can be introduced to a charging session of each EV. If the EV responds properly to this pause, paused and delayed charging can be applied to it. However, this method increases system complexity and could lead to user discomfort if the EV does not respond properly to the pause.

Finally, consumer demand may compel manufacturers to address the paused and delayed charging problems. EV users could become increasingly aware that the cost-saving benefits of smart charging are diminished if their EV model is unable to deal with charging pauses. This may influence their decision when purchasing a new model, incentivizing manufacturers to resolve these issues.

## Discussion

EV smart charging is widely acknowledged for its potential to reduce charging costs and address grid-related issues. However, it is lesser known that its potential is currently limited by technical problems related to EV smart charging. In this work, we shed light on these problems by presenting the results of large-scale EV technical charging tests and by conducting model simulations to quantify the impact of the technical limitations that are currently in place for smart charging on its effectiveness. The results of large-scale EV technical analyses showed that around one-third of the tested EV models cannot handle longer charging pauses. To prevent the EVs from shifting to sleep mode, they should continuously be charged with a minimum charging current of 6 amperes after connecting to the charging station. Model simulations showed that the potential to reduce charging costs, mitigate grid congestion and offer flexibility services using smart charging is approximately halved when charging pauses cannot be considered.

Although this research indicated that it is important that EVs are able to deal with paused and delayed charging, it should be acknowledged that actual implementation of paused and delayed charging could trigger range-anxiety issues among EV users. When scheduling the charging of an EV, its departure time from the charging station has to be estimated through user input and/or by applying forecasting methods. If an EV departs from the charging station before the anticipated departure time and the vehicle has continuously been charged with a charging current of at least 6 amperes, it is ensured that the EV has at least partly been charged. However, with the application of paused and delayed charging, there is a risk that the EV will receive a minimal charge if it departs before the expected departure time. Therefore, smart charging operators must exercise caution regarding the uncertainties in their models when employing paused and delayed charging scheduling. The reader should bear in mind that this additional uncertainty was not considered in this work's model simulations for paused and delayed charging. Nevertheless, it should be realized that this real-world challenge may be largely mitigated by actively requesting user information about their charging sessions (e.g., expected departure time & charging demand). This could be achieved, for instance, through a mobile application (e.g., refs. 42,43), either by setting user-defined defaults with opt-outs or by requesting per-session preferences.

To mitigate the problem of range anxiety among EV users when implementing smart charging solutions, various local governments and municipalities have imposed minimum charging current requirements for public charging stations within their jurisdiction[44], eliminating the option to implement charging pauses. While this approach can help to alleviate range anxiety concerns, these authorities need to recognize that it significantly constrains the potential of EV smart charging.

In addition, it should be recognized that the model simulations in this research exclusively focused on quantifying the impact of the EV's inability to perform paused and delayed charging. The results of the technical smart charging tests indicated that fluctuating or intermittent charging problems also occur frequently. This could harm the roll-out of different smart charging applications, including renewable-based charging, in which the EV charging power depends on the output of a PV system or wind turbine. Implementing renewable-based charging systems has the potential to boost the self-consumption of renewable energy and enhance the integration of renewable energy technologies into the grid[22–24]. This approach helps mitigate the intermittency of renewable energy generation and reduces dependence on fossil fuels to fulfill electricity demand. For this reason, policy addressing technical charging problems for EV smart charging should also encompass the resolution of technical charging problems related to fluctuating or intermittent charging signals.

Overall, this work showed the inability of different EV models to handle charging pauses causes highly inefficient operation of smart charging, resulting in unnecessary and costly grid reinforcements and a considerable increase in charging costs. Despite the major impact of the paused and delayed charging problems, no binding legislation is currently in place to eliminate this issue. If the public sector considers these problems to be problematic, legislation that sets a minimum technical charging performance for EV models should be introduced.

## Methods
### Technical smart charging tests
The Testlab of ElaadNL in Arnhem, the Netherlands, invites EV manufacturers to test the technical charging performance of their products in their lab. All EV models undergo the same standardized charging procedure, which consists of four tests:

1. Interoperability tests: Assesses whether the tested EV model is able to charge at different charging station models;
2. Power quality emission tests: Assesses whether the charging of the tested EV model causes disturbances in the grid voltage;
3. Power quality immunity tests: Assesses whether the tested EV model can cope with fluctuations and disturbances of the grid voltage;
4. Smart charging tests: Assesses whether the tested EV model responds to different smart charging profiles.

The manufacturers are informed of the test results, which they can utilize to enhance the technical charging performance of their products.

A large majority of the sold EV models (both PHEV and BEV models) in the Netherlands have undergone the technical charging test procedure at the Testlab. This study reported the results of the technical smart charging tests that were conducted at the Testlab between 1 June 2020 and 1 January 2023. In this timeframe, 52 EV models have undergone the fluctuating charging test, 43 models have undergone the intermittent charging test and 42 models have undergone the 20-min paused and delayed charging tests. The 6-h delayed and paused charging tests have only been introduced since April 2021. Hence, the number of EV models that have undergone this test is lower: 21 models have undergone these charging tests.

A charging test was considered unsuccessful if the EV did not continue to charge when exposed to the tested charging profile or if the charging current was at least 0.5 amperes higher than the communicated charging current in the charging signal. It should be noted that the EV manufacturers could have used the test results to resolve any technical charging issues with their model.

## Model simulations - charging models

Three sets of model simulations were conducted in this work, considering three different charging models: i) a charging cost minimization model, ii) a peak grid load minimization model and iii) a model to determine the flexibility volumes that can be offered to grid operators. Each charging model will be outlined below.

The cost minimization model is a deterministic model that can be applied to a set of EV charging sessions to determine the theoretical minimum charging costs that can be achieved in a specific electricity market. In this work, it is used to compare the charging costs with and without considering charging pauses. The validity of this model has been confirmed through real-world application[45] and the model is formulated as follows:

$$\min_{P_{ch}, \phi_{n,t}} \sum_{n=0}^{N} \sum_{t=t_{arr,n}}^{t_{dep,n}} c_t P_{ch,n,t} \Delta t \tag{1a}$$

$$\text{s.t.} \quad \sum_{t=t_{arr,n}}^{t_{dep,n}} P_{ch,t,n} \Delta t = E_{dem,n} \ \forall n \tag{1b}$$

$$0 \leq P_{ch,n,t} \leq \phi_{n,t} P_{max,n} \ \forall n, t \in \{t_{arr,n}\} \tag{1c}$$

$$\phi_{n,t} P_{min,n} \leq P_{ch,n,t} \leq \phi_{n,t} P_{max,n} \ \forall n, t \in \{t_{arr,n} + \Delta t \ldots t_{dep,n}\} \tag{1d}$$

$$\phi_{n,t-1} \geq \phi_{n,t} \ \forall n, t \in \{t_{arr,n} \ldots t_{dep,n}\} \tag{1e}$$

$$\phi_{n,t}\{0,1\} \tag{1f}$$

The objective of this optimization model in (1a) is to minimize the total charging costs of all charging sessions in the set of charging sessions $\mathcal{N}$, indexed by $n = 0 \ldots N$. In this equation, $P_{ch,n,t}$ represents the charging power in kW of charging session $n$ at time $t$, $c_t$ represents the electricity tariff at time $t$ (€/kWh), $\Delta t$ represents the timestep duration in hours and $t_{arr,n}$ and $t_{dep,n}$ represent the arrival and departure time of the considered charging session, respectively. Constraint (1b) assures that the charging demand ($E_{dem,n}$) of each charging session is met at departure. The charging power is constrained in (1c) and (1d). The minimum charging power is not considered at the first timestep after arrival (see (1c)) for each EV charging session to avoid model infeasibility, which is

caused by the fact that the charging demand of some charging sessions can not be exactly met when considering 15-min timesteps and a minimum and maximum charging power. In (1d), the binary variable $\phi_{n,t}$ makes sure that $P_{ch,n,t}$ stays between the minimum required charging power ($P_{min,n}$) and the maximum charging power ($P_{max,n}$) of the considered charging session, or is 0 otherwise. Constraint (1e) assures that once an EV stops charging, it does not restart charging later. This constraint can be neglected if charging pauses can be considered.

The peak load minimization model aims to minimize the peak transformer loading in a specific LV grid when considering a set of EV charging sessions. The nature of this model is also deterministic, assuming perfect foresight in the charging session characteristics and the non-EV load. This model can provide an understanding of the maximum potential to lower the peak transformer load when considering a given set of EV charging sessions. It is formulated as follows:

$$\min_{\substack{P_{ch}, \phi, \\ P_{grid}, P_{grid}^{peak}}} P_{grid}^{peak} \tag{2a}$$

$$\text{s.t.} \quad P_{grid,t} = P_{non-EV,t} + \sum_{n=0}^{N} P_{ch,n,t} \ \forall t \tag{2b}$$

$$P_{grid,t} \leq P_{grid}^{peak} \ \forall t \tag{2c}$$

$$(1b) - (1f) \tag{2d}$$

The objective of this model in (2a) is to minimize the peak transformer loading of the transformer ($P_{grid}^{peak}$). In (2b), $P_{grid,t}$ represents the transformer loading at timestep $t$. This is equal to the sum of the non-EV load in the considered LV grid ($P_{non-EV,t}$) and the total charging demand of all charging sessions at the considered timestep. In (2c), it is defined that the transformer load should be lower or equal to the peak transformer load at all timesteps. Lastly, the constraints in (1b)–(1f) are considered in this model.

The last optimization model determines the available downward flexibility of an EV fleet during a specified flexibility request window. This deterministic model is based on ref. 37 and formulated as follows:

$$\max_{\substack{P_{ch}, \phi, \\ P_{ch}^{tot}, P_{flex}}} P_{flex} \tag{3a}$$

$$\text{s.t.} \quad P_{ch,t}^{tot} = \sum_{n=0}^{N} P_{ch,n,t} \ \forall t \tag{3b}$$

$$P_{flex} = P_{ch,t}^{ref} - P_{ch,t}^{tot} \ \forall t \in T_{flex} \tag{3c}$$

$$(1b) - (1f) \tag{3d}$$

This model's objective in (3a) aims to maximize the downward flexibility ($P_{flex}$) that can be offered using an EV fleet during all considered timesteps in the flexibility request window. The variable $P_{ch}^{tot}$ represents the realized aggregated charging power at timestep $t$, as visible in (3b). The constraint in (3c) defines $P_{flex}$ as the difference between the charging power with the reference charging schedule ($P_{ch}^{ref}$, exogenous model input) and the realized aggregated charging power. The reference charging power depends on the reference charging strategy, e.g. uncontrolled charging or day-ahead market optimization. Constraint (3c) only applies to the set of timesteps in the considered flexibility request window ($T_{flex}$). Lastly, this model also considers the constraints in (1b)–(1f).

## Model simulations - simulation outline

All model simulations in this work were conducted using an assessment timeframe of one year, between 1 February 2022 and 1 February 2023, considering 15-min timesteps. The charging cost optimization model was applied to the whole set of considered EV charging sessions in the assessment timeframe. The hourly day-ahead market prices for different countries in Europe were used as price inputs in this optimization model. For every considered country, the optimization model was run for the scenarios with and without paused and delayed charging. In the model simulations without paused and delayed charging, charging pauses are not considered for all charging sessions to account for the fact that the operator does not know the respective EV model in the current communication protocol[31]. For countries with multiple bidding zones, the analysis is repeated for every bidding zone and the average charging costs for all bidding zones are reported. For comparison, the charging costs are also determined for uncontrolled EV charging, in which the EVs charge with maximum charging power directly after arrival until their charging demand is met. In the model simulations, it is assumed that the charging demand of EVs with a connection time to the charging station of more than 24 h will be fulfilled within one day, by setting a virtual departure time of 24 h after the time of arrival. This is done since it is not reasonable to assume that the charging demand of EVs can be delayed over multiple days, due to the unpredictable departure times of EVs. The model simulation timeframe is one day longer than the assessment timeframe to allow EVs that arrive close to the end of the assessment timeframe to complete their charging session.

The peak load minimization model is run for a varying number of considered EV charging stations. A subset of the charging stations in the EV charging session data is randomly selected for each number of considered charging stations. The model simulations include all sessions that occurred at the selected subset of charging stations during the assessment period. This process is repeated 100 times for each considered number of charging stations. Similar to the model simulations with the charging cost optimization model, the simulations were conducted considering both the case of no charging pauses and the case that considers charging pauses, as well as uncontrolled charging. The simulations also considered a virtual departure time of 24 h after the time of arrival and a model simulation timeframe of one day longer than the assessment timeframe.

The downward flexibility model was used to determine the available downward flexibility for each hour for each day in the assessment timeframe. All charging sessions of the total charging session set that were connected to the charging station during the considered hour in the assessment timeframe were included in the model simulations. An uncontrolled charging profile was considered as the reference charging profile in these model runs. Both the cases of delayed and paused charging and no delayed and paused charging were considered in the model simulations.

All model simulations were performed in Python v3.9.12[46] and Gurobi v9.5.2[47] on the DelftBlue[48] and Eejit[49] high-performance computing (HPC) clusters.

## Model simulations - data inputs & preparation

Three data sources were considered in these simulations. Historical EV charging session data was used as input for all three simulation models. This study considered EV charging data from public charging stations of charge point operator 'We Drive Solar'. Fast chargers are not included in this charging data. In this charging data, each charging session's arrival time, departure time, car ID, charging card ID, charging station ID and charging demand (kWh) is logged. Similarly, the maximum charging power for each charging station is logged at a 10 or 20-min interval, depending on the considered charging station. The maximum charging power during each charging session has been derived from this. This maximum charging power has been considered for all timesteps in the model simulations.

All model simulations only considered charging session data from public, on-street charging stations located in the city of Utrecht, the Netherlands. These stations were accessible to both PHEVs and BEVs. Charging stations that were predominantly used by EVs in car-sharing schemes (>50% of the charging sessions were from shared EVs), that were not located in residential areas (determined using visual inspection of the charging station location) and that were not active during all months of the considered assessment period were excluded from the analysis. This resulted in EV charging data from 322 charging stations, each with 2 charging sockets.

Prior to running the model simulations, the EV charging session data underwent several data preparation steps to address any data logging errors. Charging sessions that were infeasible due to data errors (i.e., the charging demand that cannot be met with the logged maximum charging power during the connection timeframe) were removed from the data. Similarly, charging sessions with a charging demand of less than 1 kWh, a maximum charging power of less than 0 kW or more than 23 kW or a connection time to the charging station of less than 15 min were omitted from the charging session data. Some charging sessions in the data had exactly the same arrival time (to the nearest second) and were registered at the same charging station ID. Due to the small probability of this occurring, these charging sessions were identified as erroneous. If the charging sessions with the same arrival time and charging station ID also had the same charging card ID, the first charging session was kept. Otherwise, both charging sessions with identical arrival times and charging station IDs were removed. The arrival and departure time of all charging sessions was rounded down to the previous 15-min timestep. For the few sessions that became infeasible due to this rounding, the charging volume was set equal to the maximum possible charging volume in the adjusted connection time to the charging station. On average, the volume of these sessions changed by 0.6 kWh. Out of all the charging sessions, 2.7% were eliminated during the data preparation process, leaving 179,374 sessions in the considered assessment timeframe.

The maximum charging power of each session was used to determine whether the EV was charging using one or three phases. EVs with maximum charging power below 7.5 kW were classified as one-phase, while all other EVs were classified as three-phase. With a minimum required charging current of 6 amperes, the minimum charging power of EVs classified as one-phase equals 1.38 kW (1 phase × 0.23 kV × 6A). The minimum charging power for three-phase EVs equals 4.14 kW (3 phases × 0.23 kV × 6A). For a low number of charging sessions (0.4%), the minimum charging power of a charging session exceeds its maximum charging power. For those sessions, the minimum charging power is set as equal to its maximum charging power to avoid model infeasibility.

The cost-minimization model also considered day-ahead electricity price data. This data was obtained from ref. 50. Transformer load data was used as input for the peak load minimization model. This study used transformer load data from one LV transformer located in a residential area in the city of Utrecht, the Netherlands. The transformer has a capacity of 400 kW and the transformer load was measured at a 15-min resolution. The non-EV loading at each timestep was determined by subtracting the loading of the registered charging stations connected to the transformed from the measured transformer loading. The peak non-EV transformer loading during the considered assessment timeframe equalled 314.5 kW.

## Reporting summary

Further information on research design is available in the Nature Portfolio Reporting Summary linked to this article.

## Data availability

Source data for Figs. 2 and 4-6 are available from https://doi.org/10.5281/zenodo.10932795[51]. The EV charging session data from 'We Drive Solar' and transformer load data from 'Stedin' that served as input for the model simulations are not publicly available due to the inclusion of privacy-sensitive information and their commercial value for these organizations. Sample data for these data inputs are provided in the data repository that was created for this work[51]. Full data access can be requested from the corresponding author. Price data that were used as inputs for the model simulations are publicly available from ref. 50. Source data are provided with this paper.

## Code availability

The code used to conduct model simulations in this work is available from https://doi.org/10.5281/zenodo.10932829[51].

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

## Acknowledgements

This study was supported by the Topsector Energy subsidy scheme of the Dutch Ministry of Economic Affairs and Climate Policy through the project 'Slim laden met flexibele nettarieven in Utrecht (FLEET)' under grand agreement TEUE519004 (N.B, P.M, B.v.d.R., H.F., B.d.B, W.v.S.), by the Dutch Ministry of Economic Affairs and Climate Policy and the Dutch Ministry of the Interior and Kingdom Relations through the ROBUST project under grant agreement MOOI32014 (N.B, N.K.P., P.M, B.v.d.R., B.d.B, S.T., W.v.S.), and by the European Union's Horizon Europe Research and Innovation program through the SCALE project under grant number 101056874 (N.B., B.v.d.R., W.v.S.).

## Author contributions

All authors conceptualized the study. N.B. guided the project. T.v.W. organized the technical smart charging tests. N.B. and N.K.P. conducted the model simulations. A.B., T.v.W. and N.K.P. gathered data and offered inputs and concepts for N.B. to write the paper. T.v.W., A.B., N.K.P., J.M., P.M., B.v.d.R., H.F., B.d.B., S.T., T.A. and W.v.S. provided feedback and input throughout the project.

## Competing interests

The authors declare no competing interests.
