## [Peer Review File · Nature Communications]

REVIEWER COMMENTS

Reviewer #1 (Remarks to the Author):

-justify the selection of the currents given in Fig. 1. why these current ranges are selected.

-Is the aim of this work only raising awareness? as it is written in the last paragraph of intro. What about solutions regarding these challenges? Some are already given. Hence, elaborate on the contribution more in this section.

-EV can handle 6 and 32 amperes. Is it for all type of EVs? What about the size of EVs? There is a need to conduct sensitivity analysis.

-When there is minimum charging current; what about charging time optimization? Charging speed is critical for some customers, not having time. Fast charging etc. Hence, discussion on the charging times and its effect on the customer satisfaction should be provided.

-Based on the conducted simulation, I suggest to add few bullet points for policy making in charging.

-Is it possible to validate the model in real environment?

-Add discussion on renewable based EV charging since the timings will be dependent on availability of solar, wind etc.

Reviewer #3 (Remarks to the Author):

This paper investigates the minimum charging power issue that may lock the potential of smart charging in reducing cost, mitigating grid congestion, and offering flexibility services. To prevent EVs from shifting to sleep mode when charging is interrupted, a minimum charging power is necessary, even at moments when a charging pause is desirable. This paper uses extensive model simulations to show the impact of this technical issue on the benefits of smart charging in the above scenarios. A quantitative analysis is conducted from the real-world data, showing that almost half of the potential benefits are lost. This paper aims to raise awareness of this technical issue and also proposes several legal and practical solutions.

In general, this paper focuses on a small problem yet shows that its impact could be surprisingly huge. The three main plots from running different charging models on the real-world data are informative and offer nice insights. However, I have major concerns about the technical contributions of this work. First, the minimum charging power is easy to model and incorporate into most of the EV charging literature. Therefore, technically it is not a novel problem. Second, the three charging models are very standard

offline formulations to schedule EV charging. However, they are not directly implementable in an online fashion in practice. Thus, the computed reduction in potential benefits is, at most, a lower bound. It may not reflect the actual impact of the minimum charging power. Last, the discussion of the options for eliminating this charging problem is a bit abstract. The enforcement methods from governments/legislation, charging operators, and consumers are briefly mentioned, but details are missing.

I also have a few other comments below.

1. The authors highlighted delayed charging problems in the paper. However, given Figure 1, it is a bit confusing since the "delay charging" test seems the least relevant (no pause or intermittency). I suggest using a more clear terminology.

2. From the smart charging test results (Figure 2), what is the percentage of unsuccessful tests that were caused by charging currents exceeding charging signals? This unsuccessful label does not seem relevant to the necessity of minimum charging power.

3. The three charging models are also stylized. If the authors consider discrete charging power instead of continuous one, I suspect the results would be much different.

Reviewer #2 (Remarks to the Author):

Please see the attached comments / suggested edits / queries on the PDF.

I think the Abstract is fairly clear - but not fully so.

Whose awareness should be raised, and with what 'theory of change' for impact?

In the Introduction:

Really needs to say more about peaks for which type of EV charging context ['on-street', home, work-place'].

Is it just EV owners, or (also) users - and how do you see the distinction between the two in your article / argument?

You really need to give the reader a definition of 'smart charging'.

Is there a pattern of EV models to which your concerns apply: Hybrid EVs? Full EVs? Depending on EV battery capacity?

How do you mean by having paid little attention? Researchers? Regulators? Governments? Or perhaps [less likely, and if so more deliberate?] EV manufacturers?

Current legislation? Do you mean standards / regulation / legislation?

At global/international level [IEC], EU?, national, regional/local?

Results:

see Figure 2: Is there a pattern of EV models to which your concerns apply: Hybrid EVs? Full EVs? Depending on EV battery capacity?

2.2.1. You should have mentioned mentioned [both] Time-of-Use and Dynamic Electricity [EV charging] Prices much earlier before - even in the Abstract.

2.2.2. I don't follow with the logical connections that "grid congestion solutions will fail with lower EV adoption rates". Why / how so?

Figure 4: Which LV grid A Dutch one? [where?]

2.2.3. Look at the grammar in the first sentence.

2.3 Needs a bit re-writing in more precise language - see my queries on the PDF.

Figure 5: which types of charging stations, and where?

Will most EV models "soon" be able to perform V2G / bi-directional charging?

Think a bit more about at which level [EU / national] these potential voluntary agreements with manufacturers?

Now you are talking about EV "buyers", previously "owners and at times "users". Work this through and clarify this right through. And a significantly share of EVs is now leased ...

Be clearer about where EV charging costs are constituted / come about. Conceptually, perhaps this might stimulate thinking: <https://www.sciencedirect.com/science/article/pii/S0306261919316526>

Unlocking the full potential of smart charging: Addressing delayed charging problems in electric vehicles

Nico Brinkel^{1,*}, Thijs van Wijk², Anoeska Buijze³, Nanda Kishor Panda⁴, Jelle Meersmans⁵, Peter Markotić², Bart van der Ree⁶, Henk Fidler⁷, Baerte de Brey^{2,7}, Simon Tindemans⁴, Tarek AISkaif⁸ and Wilfried van Sark¹

¹Copernicus Institute of Sustainable Development, Utrecht University, Princetonlaan 8a, 3584 CB Utrecht, the Netherlands

²ElaadNL, Westervoortsedijk 73, 6827 AV Arnhem, the Netherlands

³Faculty of Law, Economics and Governance, Utrecht Centre for Water, Oceans and Sustainability Law, Utrecht University, Newtonlaan 201, 3584 BH Utrecht, the Netherlands

⁴Department of Electrical Sustainable Energy, Delft University of Technology, Mekelweg 4, 2628 CD Delft, the Netherlands

⁵Enervalis, Lummense Kiezel 51, 3500 Hasselt, Belgium

⁶Utrecht Sustainability Institute, Postbus 85057, 3508 AB Utrecht, the Netherlands

⁷Stedin Groep, Blaak 8, 3011 TA Rotterdam, the Netherlands

⁸Information Technology Group (INF), Wageningen University and Research (WUR), 6706 KN Wageningen, the Netherlands

* Corresponding author: n.b.g.brinkel@uu.nl

Smart charging of electric vehicles can help tackle grid congestion and reduce charging costs. However, it is not widely known that the smart charging potential is currently limited by the technical capabilities of electric vehicles. In particular, many electric vehicle models are unable to deal with charging pauses or delays. These models shift to sleep mode when charging is interrupted and charging cannot be resumed after the interruption. To avoid this, they should be continuously charged with a minimum charging power, even at moments when a charging pause would be desirable. This research examines this problem to increase awareness and initiate steps to resolve it. Through technical charging tests, it was found that around one-third of tested car models suffer from this issue. Model simulations showed that eliminating this minimum charging power requirement would double the potential of smart charging from both a cost-reduction and a grid congestion mitigation perspective. Lastly, concrete legal and practical solutions are proposed for eliminating the delayed charging problem.

1 Introduction

With the large-scale adoption of Electric Vehicles (EVs), our road transportation and electricity systems become increasingly intertwined, where the energy requirements to fulfill our road transportation needs were initially provided by a network of petrol stations, this role is increasingly taken over

by the electricity grid infrastructure through EV charging [1, 2].

This transition brings new challenges to the electricity system, particularly to the grid infrastructure. Most EV charging occurs in Low-Voltage (LV) grids and the majority of these grids were designed decades ago without the concept of EV charging in mind. The charging power of an EV is significantly higher than the average power consumption of a household, and since most EVs tend to arrive at their charging stations at a similar time, concentrated charging moments are expected [3, 4]. As a result, EV charging is likely to cause grid congestion [5–7].

Grid reinforcements could serve as a solution. However, from an economic and practical standpoint, large-scale reinforcements are infeasible due to their high costs [8] (>100,000 €/km for LV cables in the Netherlands [9]) and the shortage of personnel with technical expertise to carry out these reinforcements [10, 11].

Instead, grid congestion problems can also be mitigated by stepping away from *uncontrolled charging*, in which EVs charge at maximum power directly after arrival until their charging demand is met. For most EV charging transactions, the connection time to a charging station considerably exceeds the required time to meet their charging demand. This provides ample opportunities to shift the charging demand over time using EV smart charging [12, 13].

EV smart charging is commonly seen as something that can benefit both grid operators and EV owners. Different studies showed that the application of smart charging can support grid operators in mitigating grid congestion and power qual-

Figure 1: Overview of the charging currents for the different smart charging tests in the charging protocol at the ElaadNL Testlab.

ity problems (e.g., [14–16]). Similarly, smart charging can be applied for the provision of balancing reserves to Transmission System Operators (TSOs) [17, 18]. Additionally, smart charging can help EV owners to reduce their charging costs by taking advantage of low electricity market prices [19, 20].

The current deployment of smart charging is hindered by the technical capabilities of EVs. A significant share of EV models is unable to perform paused or delayed charging, in which the charging process is paused for a longer time [21]. A charging pause causes these EV models to switch to sleep mode and makes them unresponsive to charging signals after this pause, posing a risk of unmet charging demand at their departure from the charging station. To avoid this, EVs need to be charged continuously with at least their minimum charging power [22], even when this is not desirable (e.g., moments with a high price or grid load).

Remarkably, the technical problems associated with smart charging have received little attention, leading to low awareness about these issues. This is evident from several factors. First, almost all smart charging studies fail to consider that a considerable share of the EV fleet cannot perform delayed charging, resulting in an overestimation of the smart charging potential. Second, a notable portion of newly-introduced EV models cannot perform delayed charging, indicating that manufacturers may not be aware of this issue. Third, there appear to be no legal or policy initiatives addressing this problem.

This work aims to shed light on these problems, in order to raise awareness about the technical limitations associated with EV charging and to initiate steps toward eliminating them. To provide better insight into the frequency of occurrence of these problems, results of large-scale technical charging tests are presented. Moreover, this study presents results of model simulations that quantify the impact of the required minimum charging current on three different services which can be provided using smart charging: i) charging

cost reduction, ii) mitigation of grid congestion, and iii) offering flexibility products to grid operators. Lastly, current legislation on this topic is discussed and options to eliminate delayed charging problems are discovered.

2 Results

2.1 Technical Smart Charging Test

The technical performance of new EV and charging station models is evaluated using charging tests at the Testlab of ElaadNL, a knowledge center on EV charging founded by Dutch grid operators. Manufacturers of EVs and charging stations are invited to test the technical performance of their products. They are tested on their interoperability, their impact on power quality and their ability to perform smart charging [23]. The results can be used by manufacturers to improve the technical performance of their products. A large share of the EV models on the Dutch market have undergone charging tests at the Testlab. This section reports the smart charging test results, but it is important to note that some manufacturers have since rectified the charging issues through the installation of software updates.

The smart charging tests aim to evaluate the EV's response when exposed to various charging signals (see Fig. 1). The first test assesses whether an EV can handle a fluctuating charging signal between 6 and 32 amperes, with a 60-second interval. The intermittent charging test evaluates the EV's response to a fluctuating charging signal between 0 and 6 amperes, which is repeated 25 times with a 60-second interval. The last set of tests analyzes the EV's response to paused and delayed charging. For the paused charging tests, the EV is charged for a brief period, followed by a pause in the charging session. The delayed charging tests assess the EV's ability to properly react to the charging signal after the

Figure 2: Results of the conducted charging tests at the ElaadNL Test-lab.

charging is delayed from the start of the transaction. Both tests are conducted with pauses of 20 minutes and 6 hours.

The results of the smart charging tests are presented in Figure 2. A charging test was labeled as unsuccessful if the tested EV model ceased charging or if the charging current of the EV exceeded the charging signal by at least 0.5 amperes. The success rate of the fluctuating charging test equals 71% for the tested EV models. The share of tested models that can follow the intermittent charging profile is lower and equals 63%. This can be a barrier to a successful roll-out of load balancing at parking lots (i.e., quickly alternating the charging power between charging stations to reduce the peak charging power of the parking lot) and of solar charging (i.e., only charging at moments with solar generation), due to the rapid fluctuations in charging power that can occur with these charging modes. These issues appear to be caused by the software settings of certain EV models, which identify charging stations with high fluctuations in the charging signal as faulty, without considering that these fluctuations may be caused by the application of smart charging.

The share of EVs that are able to deal with delayed and paused charging profiles is dependent on the duration of the pause. When considering a charging pause of 20 minutes, the success rate for the tested EV models equals 86% and 83% for paused and delayed charging, respectively. When the duration of the pause is extended to 6 hours, the share of tested EV models that successfully pass the charging test reduces to 71% and 67% for paused and delayed charging, respectively. These issues are also software-based: to prevent the 12-volt battery that powers the vehicle's electrical systems from draining, the EVs switch to sleep mode if no charging signal is received for an extended duration.

2.2 Impact on Smart Charging Potential

The inability of different EV models to handle charging pauses complicates the application of smart charging for different functions. While longer charging pauses may be desirable at specific moments, for instance with high grid load, they cannot be implemented into EV charging schedules as some EV models will shift into pause mode. The only way to reduce the impact of EV charging at these moments is by charging with the lowest possible current of the charging system (typically

6 amperes). Since the EV model is not specified in the current communication protocols between the EV, the charging station and the back-office of the charge point operator [24, 25], a minimum charging current is generally considered for all EVs, regardless of whether they are capable of performing delayed charging or not.

The required minimum charging current can diminish the effectiveness of different smart charging applications. On the one hand, EVs are forced to charge with the minimum current when they ideally would not (e.g., when the grid is congested for consumption). On the other hand, EVs cannot always take advantage of beneficial charging times (e.g., with low electricity prices), since their charging demand has already been met by charging with the minimum charging current at earlier moments. Model simulations are performed to quantify the impact of the required minimum charging current on three different services which can be provided using smart charging: i) charging cost reduction, ii) mitigation of grid congestion and iii) offering flexibility products to grid operators.

2.2.1 Charging cost reduction

First, smart charging offers the potential to reduce charging costs when participating in an electricity market that considers dynamic pricing (e.g., day-ahead market for electricity) [26–28]. This is done by shifting the charging demand from moments with high prices to moments with low prices. Fig. 3 presents the charging costs for a large EV fleet when participating in the day-ahead market in different European countries with perfect foresight. Three charging scenarios are considered in the model simulations: i) cost-optimization without considering a minimum charging current, ii) cost-optimization with a minimum charging current of 6 amperes, and iii) uncontrolled charging. The results in Fig. 3 indicate that the required minimum charging current almost halves the cost-reduction potential of smart charging. When no minimum charging current is considered, the cost-reduction potential compared to uncontrolled charging equals 14-35%, depending on the country. For all considered countries, the cost reduction potential is approximately half as high when a 6-ampere minimum charging current is considered. This is because EVs are forced to charge with the minimum current at times of high prices and cannot benefit from lower prices since their charging demand is fulfilled before those times arrive.

2.2.2 Grid congestion mitigation

Smart charging can also be used to mitigate grid congestion by shifting the charging from moments with high grid load to moments with low grid load [29, 30]. When a minimum charging current needs to be applied, EV charging cannot be completely shifted away from peak hours. Hence, grid congestion solutions will fail with lower EV adoption rates.

To investigate the impact of the minimum required charging current on the potential for mitigating grid congestion through smart charging, model simulations were conducted using a transformer peak load minimization algorithm for EV

Figure 3: Average EV charging costs for three different charging strategies in the day-ahead market of different European countries. Percentage values represent the cost decrease compared to uncontrolled charging (blue bars). For countries with multiple bidding zones (DK, NO, SE & IT), the average charging costs for all bidding zones are reported. As Germany and Luxembourg comprise one bidding zone, the results for these countries are reported together.

Figure 4: Peak transformer load values in a one-year assessment time-frame when applying a peak transformer load minimization algorithm to a different number of installed charging stations in **one LV grid**. Analysis is repeated 100 times for each considered number of EV charging stations, with a randomly sampled subset of EV charging stations in each run. The line shows the average outcome for all model runs and the shaded area shows the 95% confidence interval.

charging. Simulations were performed for both no minimum required charging current and a minimum charging current of 6 amperes.

Fig. 4 presents the transformer peak load values (i.e., sum of the non-EV load and EV load) for a varying number of charging stations connected to one LV grid. From this figure, one can determine the maximum number of EV charging stations that can be theoretically hosted in one LV grid without causing transformer congestion. The results show that **the number of charging stations** that can be installed in the considered grid without exceeding the transformer capacity is approximately twice as high when the minimum charging current does not need to be considered. Transformer congestion problems will occur in the studied grid if it hosts between 35-55 charging stations, a peak load minimization algorithm is applied and a minimum charging current is considered. When neglecting the minimum required charging current, between 75-115 charging stations can be hosted in the same grid.

2.2.3 Offering of flexibility products

Moreover, smart charging can be applied to offer flexibility services to grid operators, such as supplying balancing reserves (e.g., frequency restoration reserves) to TSOs to restore the balance between supply and demand [31–33]. Additionally, DSOs have been experimenting with local flexibility markets to address grid congestion issues [34]. The minimum required charging current affects the amount of downward flexibility (i.e., a reduction in charging power from the reference charging schedule) that can be provided using smart charging. This was demonstrated in the final series of model simulations, which compared the available downward flexibility of an EV fleet under two scenarios: one that

considers the minimum charging current requirement, and another that does not. In these simulations, the maximum reduction in charging power from the reference charging schedule (uncontrolled charging in this case) is determined, while ensuring that the charging demand of every transaction is met at departure.

The violin plot in Fig. 5 presents the distribution of the hourly available downward flexibility throughout one year for both considered cases. It shows that the available downward flexibility is higher during early evening hours. This is because more EVs are typically charging during this time in residential areas, resulting in higher potential for charging power reduction. As visible in Fig. 5, the minimum required charging current reduces the available downward flexibility, since EVs that consider this minimum charging current cannot fully ramp down their charging power to provide downward flexibility and must keep the charging current above the minimum charging current. On average, 34% less downward flexibility can be offered when considering a minimum charging current of 6 amperes (95% CI: 24-44%).

2.3 Options for eliminating charging problems

The previous section emphasized the necessity of EV car models to be able to perform delayed charging. This section examines **existing legislation** on this issue and explores solutions to avoid such problems in the future.

Two standards in the **current EU legislation** address the delayed and paused charging problems. However, EV manufacturers are not obliged to comply with them. IEC 61851 [25] is a set of **standards** that contains use cases for EVs and charging stations on how to wake up EVs that shifted to sleep mode after a charging pause. This standard can be used by manufacturers of EVs and charging stations to comply with **the Low Voltage Directive (LVD)** [35]. This directive applies to all electrical equipment traded within the EU and aims to safeguard the health and safety of persons, animals and property (art. 1 LVD). It is based on **self-assessment** and there is no notified body that intervenes with the conformity assessment procedure. For EVs and charging stations, compliance with IEC 61851 leads to the presumption of compliance with the LVD (art. 12 LVD). By complying with this standard, EVs are able to perform delayed charging. However, EV manufacturers do not need to use this standard to comply with the LVD; as long as the safety standards in the LVD are met, the product can be traded on the EU market. Since the ability to handle charging pauses is of limited relevance for product safety, EVs are able to comply with the LVD without complying with IEC 61851.

Secondly, the delayed charging problem is addressed in a **newly-proposed** standard for communication between EVs and their charging station. The ISO 15118-20 standard [36] has been developed to enable bidirectional EV charging through vehicle-to-grid (V2G) technology and has been implemented in a small number of EV models. It includes a use case to re-establish communication with an EV in sleep mode by sending a wake-up trigger. While the implementation of

Figure 5: Violin plot with the distribution of the available downward flexibility of an EV fleet for each hour of the day during the assessment timeframe of one year when considering the uncontrolled charging profile as the reference charging profile. Results are compared for the case with and without a considered minimum required charging current. Model simulations are performed using historical charging data from 322 charging stations. Stripes show the 25%, 50% and 75% quantile values.

this standard would likely eliminate the delayed charging problem for compliant EVs, there is no legal requirement for manufacturers to incorporate it in their models. As a result, the delayed charging problem might persist for some EV models, since manufacturers might not be incentivized to implement the standard in models that are unable to perform V2G functions.

If governments deem that the delayed charging problems are too severe to be resolved without intervention, there are multiple enforcement methods available. First, they could eliminate this problem through voluntary agreements with manufacturers. Governments can also enforce this by implementing regulations on this topic. This could be modeled after the type-approval system that is currently in place. Before getting access to public roads in the EU, all car models need to acquire type-approval [37], issued by a national approval authority. The type-approval tests assess whether the car models meet EU safety rules (e.g., crash tests), as well as noise and emission limits. Similarly, EU legislation could make compliance with for instance ISO 15118-20 compulsory, either through self-assessment or by requiring testing and approval by a national approval authority.

As long as the delayed charging issues are not resolved, smart charging operators can use workaround solutions to identify whether a specific EV can handle charging pauses. A pause can be introduced to a charging session of each EV. If the EV responds properly to this pause, delayed charging can be applied to it. However, this method increases system complexity and could lead to user discomfort if the EV does not respond properly to the pause.

Finally, consumer demand may compel manufacturers to address the delayed charging problems. EV buyers could be

come increasingly aware that the cost-saving benefits of smart charging are diminished if a minimum charging current is required. This may influence their decision when purchasing a new model, incentivizing manufacturers to resolve these issues.

3 Conclusions

EV smart charging can potentially reduce charging cost and mitigate grid issues. However, it is generally neglected that its potential is diminished by technical charging problems of EVs, in particular the inability of different EV models to perform delayed or paused charging. The results of large-scale EV technical analyses showed that around one-third of the tested EV models cannot handle longer charging pauses. To prevent the EVs from shifting to sleep mode, they should be charged with a minimum charging current of 6 amperes. This has a major impact on the potential benefits that can be obtained using smart charging. Model simulations showed that the potential to reduce charging costs, mitigate grid congestion and offer flexibility services using smart charging is approximately halved when the minimum charging current needs to be considered. Hence, the required minimum charging current causes highly inefficient operation of smart charging, resulting in unnecessary and costly grid reinforcements and a considerable increase in charging costs. Despite the major impact of the required minimum charging current, there is currently no binding legislation in place to eliminate this issue. If governments consider the delayed charging issues to be problematic, legislation that sets a minimum technical charging performance for EV models should be introduced.

A Methods

A.1 EV charging tests

The Testlab of ElaadNL in Arnhem, the Netherlands, invites EV manufacturers to test the technical charging performance of their products in their lab. All EV models undergo the same standardized charging procedure, which consists of four tests:

1. Interoperability tests: Assesses whether the tested EV model is able to charge at different charging station models;
2. Power quality emission tests: Assesses whether the charging of the tested EV model causes disturbances in the grid voltage;
3. Power quality immunity tests: Assesses whether the tested EV model can cope with fluctuations and disturbances of the grid voltage;
4. Smart charging tests: Assesses whether the tested EV model responds to different smart charging profiles.

The manufacturers are informed of the test results, which they can utilize to enhance the technical charging performance of their products.

A **large majority** of the sold EV car types in the Netherlands have undergone the technical charging test procedure at the Testlab. This study reported the results of the technical smart charging tests that are conducted at the Testlab between 1 June 2020 and 1 January 2023. In this timeframe, 52 EV models have undergone the fluctuating charging test, 43 models have undergone the intermittent charging test and 42 models have undergone the 20-minute paused and delayed charging tests. The 6-hour delayed and paused charging tests have only been introduced since April 2021. Hence, the number of EV models that have undergone this test is lower: 21 models have undergone these charging tests.

A charging test was considered unsuccessful if the EV did not continue to charge when exposed to the tested charging profile, or if the charging current was at least 0.5 amperes higher than the communicated charging current in the charging signal. It should be noted that the EV manufacturers could have used the test results to resolve any technical charging issues with their model.

A.2 Model simulations

A.2.1 Charging models

Three sets of model simulations were conducted in this work, considering three different charging models: i) a charging cost minimization model, ii) a peak grid load minimization model and iii) a model to determine the flexibility volumes that can be offered to grid operators. Each charging model will be outlined below.

The cost minimization model is a deterministic model that can be applied to a set of EV charging transactions to determine the theoretical minimum charging costs that can be achieved in a specific electricity market. In this work, it is used to compare the charging costs with and without the

minimum charging current requirement. It is formulated as follows:

$$\min_{P_{ch}, \phi_{n,t}} \sum_{n=0}^N \sum_{t=t_{arr,n}}^{t_{dep,n}} c_t P_{ch,n,t} \Delta t \quad (1a)$$

$$\text{s.t.} \quad \sum_{t=t_{arr,n}}^{t_{dep,n}} P_{ch,t,n} \Delta t = E_{dem,n} \quad \forall n \quad (1b)$$

$$0 \leq P_{ch,n,t} \leq \phi_{n,t} P_{max,n} \quad \forall n, t \in \{t_{arr,n}\} \quad (1c)$$

$$\phi_{n,t} P_{min,n} \leq P_{ch,n,t} \leq \phi_{n,t} P_{max,n} \quad \forall n, t \in \{t_{arr,n} + \Delta t \dots t_{dep,n}\} \quad (1d)$$

$$\phi_{n,t-1} \geq \phi_{n,t} \quad \forall n, t \in \{t_{arr,n} \dots t_{dep,n}\} \quad (1e)$$

$$\phi_{n,t} \in \{0, 1\} \quad (1f)$$

The objective of this optimization model in (1a) is to minimize the total charging costs of all charging transactions in the set of charging transactions \mathcal{N} , indexed by $n = 0 \dots N$. In this equation, $P_{ch,n,t}$ represents the charging power in kW of charging transaction n at time t , c_t represents the electricity tariff at time t (€/kWh), Δt represents the timestep duration in hours and $t_{arr,n}$ and $t_{dep,n}$ represent the arrival and departure time of the considered charging transaction, respectively. Constraint (1b) assures that the charging demand ($E_{dem,n}$) of each charging transaction is met at departure. The charging power is constrained in (1c) and (1d). The minimum charging power is not considered at the first timestep after arrival (see (1c)) for each EV charging transaction to avoid model infeasibility, which is caused by the fact that the charging demand of some charging transactions can not be exactly met when considering 15-minute timesteps and a minimum and maximum charging power. In (1d), the binary variable $\phi_{n,t}$ makes sure that $P_{ch,n,t}$ stays between the minimum required charging power ($P_{min,n}$) and the maximum charging power ($P_{max,n}$) of the considered charging transaction, or is 0 otherwise. Constraint (1e) assures that once an EV stops charging, it does not restart charging at a later point in time. This constraint can be neglected if no minimum charging current is considered.

The peak load minimization model aims to minimize the peak transformer loading in a specific LV grid when considering a set of EV charging transactions. The nature of this model is also deterministic, assuming perfect foresight in the charging transaction characteristics and the non-EV load. This model can provide an understanding of the maximum potential to lower the peak transformer load when considering a given set of EV charging transactions. It is formulated as follows:

$$\min_{P_{ch}, \phi, P_{grid}, P_{grid}^{peak}} P_{grid}^{peak} \quad (2a)$$

$$\text{s.t.} \quad P_{grid,t} = P_{non-EV,t} + \sum_{n=0}^N P_{ch,n,t} \quad \forall t \quad (2b)$$

$$P_{grid,t} \leq P_{grid}^{peak} \quad \forall t \quad (2c)$$

$$(1b) - (1f)$$

The objective of this model in (2a) is to minimize the

peak transformer loading of the transformer ($P_{\text{grid}}^{\text{peak}}$). In (2b), $P_{\text{grid},t}$ represents the transformer loading at timestep t . This is equal to the sum of the non-EV load in the considered LV grid ($P_{\text{non-EV},t}$) and the total charging demand of all charging transactions at the considered timestep. In (2c), it is defined that the transformer load should be lower or equal to the peak transformer load at all timesteps. Lastly, the constraints in (1b)-(1f) are considered in this model.

The last optimization model determines the available downward flexibility of an EV fleet during a specified flexibility request window. This deterministic model is formulated as follows:

$$\max_{\substack{P_{\text{ch}}, \phi, \\ P_{\text{ch}}^{\text{tot}}, P_{\text{flex}}}} P_{\text{flex}} \quad (3a)$$

$$\text{s.t.} \quad P_{\text{ch},t}^{\text{tot}} = \sum_{n=0}^N P_{\text{ch},n,t} \quad \forall t \quad (3b)$$

$$P_{\text{flex}} = P_{\text{ch},t}^{\text{ref}} - P_{\text{ch},t}^{\text{tot}} \quad \forall t \in T_{\text{flex}} \quad (3c)$$

$$(1b) - (1f)$$

This model's objective in (3a) aims to maximize the downward flexibility (P_{flex}) that can be offered using an EV fleet during all considered timesteps in the flexibility request window. The variable $P_{\text{ch}}^{\text{tot}}$ represents the realized aggregated charging power at timestep t , as visible in (3b). The constraint in (3c) defines P_{flex} as the difference between the charging power with the reference charging schedule ($P_{\text{ch}}^{\text{ref}}$, exogenous model input) and the realized aggregated charging power. The reference charging power depends on the reference charging strategy, e.g. uncontrolled charging or day-ahead market optimization. Constraint (3c) only applies to the set of timesteps in the considered flexibility request window (T_{flex}). Lastly, this model also considers the constraints in (1b)-(1f).

A.2.2 Simulation outline

All model simulations in this work were conducted using an assessment timeframe of one year, between 1 February 2022 and 1 February 2023, considering 15-minute timesteps. The charging cost optimization model was applied to the whole set of considered EV charging transactions in the assessment timeframe. The hourly day-ahead market prices for different countries in Europe were used as price inputs in this optimization model. For every considered country, the optimization model was run for both a minimum charging current of 6 amperes and no minimum charging current. In the model simulations with a minimum charging current of 6 amperes, the minimum charging current is applied to all charging transactions, to account for the fact that the EV model is not known by the operator in the current communication protocol [24]. For countries with multiple bidding zones, the analysis is repeated for every bidding zone and the average charging costs for all bidding zones are reported. For comparison, the charging costs are also determined for uncontrolled EV charging, in which the EVs charge with maximum charging power directly after arrival until their charging demand is met. In the model

simulations, it is assumed that the charging demand of EVs with a connection time to the charging station of more than 24 hours will be fulfilled within one day, by setting a virtual departure time of 24 hours after the time of arrival. This is done since it is not reasonable to assume that the charging demand of EVs can be delayed over multiple days, due to the unpredictable departure times of EVs. The model simulation timeframe is one day longer than the assessment timeframe to allow EVs that arrive close to the end of the assessment timeframe to complete their charging transaction.

The peak load minimization model is run for a varying number of considered EV charging stations. For each number of considered charging stations, a subset of the charging stations in the EV transaction data is randomly selected. The model simulations include all transactions that occurred at the selected subset of charging stations during the assessment period. This process is repeated 100 times for each considered number of charging stations. Similar to the model simulations with the charging cost optimization model, the simulations were conducted considering a minimum charging current of 6 amperes and no minimum charging current, as well as uncontrolled charging. The simulations also considered a virtual departure time of 24 hours after the time of arrival and a model simulation timeframe of one day longer than the assessment timeframe.

The downward flexibility model was used to determine the available downward flexibility for each hour for each day in the assessment timeframe. All charging transactions of the total charging transaction set that were connected to the charging station during the considered hour in the assessment timeframe were included in the model simulations. An uncontrolled charging profile was considered as the reference charging profile in these model runs. Both the cases of a minimum charging current 6 amperes and no minimum charging current were considered in the model simulations.

All model simulations were performed in Python v3.9.12 [38] and Gurobi v9.5.2 [39] on the DelftBlue [40] and Eejit [41] high-performance computing (HPC) clusters.

A.2.3 Data inputs & data preparation

Three data sources were considered in these simulations. Historical EV charging transaction data was used as input for all three simulation models. This study considered EV charging data from public charging stations of charge point operator *We Drive Solar*. Fast chargers are not included in this charging data. In this charging data, the arrival time, departure time, car ID, charging card ID, charging station ID and charging demand (kWh) of each charging transaction is logged. Similarly, the maximum charging power for each charging station is logged with a 10 or 20-minute interval, depending on the considered charging station. From this, the maximum charging power during each charging transaction has been derived. This is the maximum charging power that has been considered for all timesteps in the model simulations.

All model simulations only considered charging transaction data from charging stations located in the city of Utrecht, the Netherlands. Charging stations that were predominantly used by EVs in car-sharing schemes (>50% of the charging

transactions were from shared EVs), that were not located in residential areas (determined using visual inspection of the charging station location) and that were not active during all months of the considered assessment period were excluded from the analysis. This resulted in EV charging data from **322 charging stations, each with 2 charging sockets.**

Prior to running the model simulations, the EV charging transaction data underwent several data preparation steps to address any data logging errors. Charging sessions that were infeasible due to data errors (i.e., the charging demand that cannot be met with the logged maximum charging power during the connection timeframe) were removed from the data. Similarly, charging transactions with a charging demand of less than 1 kWh, a maximum charging power of less than 0 kW or more than 23 kW or a connection time to the charging station of less than 15 minutes were omitted from the charging transaction data. Some charging transactions in the data had exactly the same arrival time (to the nearest second) and were registered at the same charging station ID. Due to the small probability of this occurring, these charging transactions were identified as erroneous. If the charging transactions with the same arrival time and charging station ID also had the same charging card ID, the first charging transaction was kept. Otherwise, both charging transactions with identical arrival times and charging station IDs were removed. The arrival and departure time of all charging transactions was rounded down to the previous 15-minute timestep. For the few transactions that became infeasible due to this rounding, the charging volume was set equal to the maximum possible charging volume in the adjusted connection time to the charging station. On average, the volume of these transactions changed by 0.6 kWh. Out of all the charging transactions, 2.7% were eliminated during the data preparation process, leaving 179,374 transactions in the considered assessment timeframe.

The maximum charging power of each transaction was used to determine whether the EV was charging using one or three phases. EVs with maximum charging power below 7.5 kW were classified as one-phase, while all other EVs were classified as three-phase. With a minimum required charging current of 6 amperes, the minimum charging power of EVs classified as one-phase equals 1.38 kW (1 phase \times 0.23 kV \times 6A). The minimum charging power for three-phase EVs equals 4.14 kW (3 phases \times 0.23 kV \times 6A). For a low number of charging transactions (0.4%), the minimum charging power of a charging transaction exceeds its maximum charging power. For those transactions, the minimum charging power is set as equal to its maximum charging power to avoid model infeasibility.

The cost-minimization model also considered day-ahead electricity price data. This data was obtained from [42]. Transformer load data was used as input for the peak load minimization model. This study used transformer load data from one LV transformer located in a residential area in the city of Utrecht, the Netherlands. The transformer has a capacity of 400 kW and the transformer load was measured with a 15-minute resolution. The non-EV loading at each timestep was determined by subtracting the loading of the registered charging stations connected to the transformed

from the measured transformer loading. The peak non-EV transformer loading during the considered assessment timeframe equaled 314.5 kW.

B Acknowledgements

This study was supported by the Topsector Energy subsidy scheme of the Dutch Ministry of Economic Affairs and Climate Policy through the project 'Slim laden met flexibele nettarieven in Utrecht (FLEET)', by the Dutch Ministry of Economic Affairs and Climate Policy and the Dutch Ministry of the Interior and Kingdom Relations through the ROBUST project under grant agreement MOOI32014 and by the European Union's Horizon Europe Research and Innovation program through the SCALE project (Grant Agreement No. 101056874).

Bibliography

1. International Energy Agency. *Global EV Outlook 2023* tech. rep. (Paris, 2023). <https://www.iea.org/reports/global-ev-outlook-2023>.
2. Archsmith, J., Muehlegger, E. & Rapson, D. S. Future Paths of Electric Vehicle Adoption in the United States : Predictable Determinants , Obstacles , and Opportunities. *Environmental and Energy Policy and the Economy* **3**, 71–110 (2022).
3. Su, J., Lie, T. T. & Zamora, R. Modelling of large-scale electric vehicles charging demand: A New Zealand case study. *Electric Power Systems Research* **167**, 171–182. ISSN: 03787796. <https://doi.org/10.1016/j.epsr.2018.10.030> (2019).
4. Sadeghianpourhamami, N., Refa, N., Strobbe, M. & Devellder, C. Quantitive analysis of electric vehicle flexibility: A data-driven approach. *International Journal of Electrical Power and Energy Systems*. ISSN: 01420615 (2018).
5. Ashfaq, M., Butt, O., Selvaraj, J. & Rahim, N. Assessment of electric vehicle charging infrastructure and its impact on the electric grid: A review. *International Journal of Green Energy* **18**, 657–686. ISSN: 15435083 (2021).
6. Rahman, S., Khan, I. A., Khan, A. A., Mallik, A. & Nadeem, M. F. Comprehensive review & impact analysis of integrating projected electric vehicle charging load to the existing low voltage distribution system. *Renewable and Sustainable Energy Reviews* **153**, 111756. ISSN: 18790690 (2022).
7. Veldman, E. & Verzijlbergh, R. A. Distribution grid impacts of smart electric vehicle charging from different perspectives. *IEEE Transactions on Smart Grid* **6**, 333–342. ISSN: 19493053 (2015).
8. Anwar, M. B. *et al.* Assessing the value of electric vehicle managed charging: A review of methodologies and results. *Energy and Environmental Science* **15**, 466–498. ISSN: 17545706 (2022).

9. Brinkel, N., Schram, W., AlSkaif, T., Lampropoulos, I. & Sark, W. v. Should we reinforce the grid? Cost and emission optimization of electric vehicle charging under different transformer limits. *Applied Energy* **276**, 115285. ISSN: 0306-2619 (2020).
10. International Energy Agency. *Renewables 2022* tech. rep. (Paris, 2022), 158. <https://www.iea.org/reports/renewables-2022>.
11. Dutch Authority for Consumers & Markets. *Harder choices needed with regard to grid expansions in order to meet objectives of the energy transition 2022*. <https://www.acm.nl/en/publications/harder-choices-needed-regard-grid-expansions-order-meet-objectives-energy-transition>.
12. Develder, C., Sadeghianpourhamami, N., Strobbe, M. & Refa, N. Quantifying flexibility in EV charging as DR potential: Analysis of two real-world data sets. *2016 IEEE International Conference on Smart Grid Communications, SmartGridComm 2016*, 600–605 (2016).
13. Sørensen, L., Lindberg, K. B., Sartori, I. & Andresen, I. Analysis of residential EV energy flexibility potential based on real-world charging reports and smart meter data. *Energy and Buildings* **241**, 110923. ISSN: 03787788. <https://doi.org/10.1016/j.enbuild.2021.110923> (2021).
14. García-Villalobos, J., Zamora, I., San Martín, J. I., Asensio, F. J. & Aperribay, V. *Plug-in electric vehicles in electric distribution networks: A review of smart charging approaches 2014*.
15. Brinkel, N., AlSkaif, T. & van Sark, W. Grid congestion mitigation in the era of shared electric vehicles. *Journal of Energy Storage* **48**, 103806. ISSN: 2352152X. <https://doi.org/10.1016/j.est.2021.103806> (2022).
16. Verbist, F., Panda, N. K., Vergara, P. P. & Palensky, P. Impact of Dynamic Tariffs for Smart EV Charging on LV Distribution Network Operation. *arXiv preprint arXiv:2306.10775*. arXiv: 2306.10775 (2023).
17. Duan, X., Hu, Z. & Song, Y. Bidding Strategies in Energy and Reserve Markets for an Aggregator of Multiple EV Fast Charging Stations with Battery Storage. *IEEE Transactions on Intelligent Transportation Systems* **22**, 471–482. ISSN: 15580016 (2021).
18. Song, M., Amelin, M., Wang, X. & Saleem, A. Planning and Operation Models for EV Sharing Community in Spot and Balancing Market. *IEEE Transactions on Smart Grid* **10**, 6248–6258. ISSN: 19493061 (2019).
19. González Vayá, M. & Andersson, G. *Optimal Bidding Strategy of a Plug-In Electric Vehicle Aggregator in Day-Ahead Electricity Markets Under Uncertainty*. *IEEE transactions on Power Systems* **30**, 2375–2385. ISSN: 10003673 (2014).
20. Zheng, Y., Yu, H., Shao, Z. & Jian, L. Day-ahead bidding strategy for electric vehicle aggregator enabling multiple agent modes in uncertain electricity markets. *Applied Energy* **280**, 115977. ISSN: 03062619. <https://doi.org/10.1016/j.apenergy.2020.115977> (2020).
21. Van Dijk, J. et al. *Smart Charging Position Paper: Minimum Load Requirement: Well-intended but a Smart Charging obstacle How one requirement limits energy flexibility and the CO2 reduction potential in e-mobility* tech. rep. (2022). <https://www.e-mobility.totalenergies.nl/media/2plijlep/totalenergies-smart-charging-position-paper-minimum-load-requirement-well-intended-but-a-smart-charging-obstacle.pdf>.
22. Nationale Agenda Laadinfrastructuur (National Charging Infrastructure Agenda). *Smart Charging Requirements (SCR)* (2021).
23. ElaadNL. *Tests - Testing Innovative Solutions 2023*. <https://elaad.nl/en/topics/tests-at-the-elaadnl-testlab/>.
24. EVRoaming Foundation. *OCPI 2.2.1* tech. rep. (2021). <https://evroaming.org/app/uploads/2021/11/OCPI-2.2.1.pdf>.
25. International Electrotechnical Commission. *IEC 61851-1:2019* 2019. <https://www.nen.nl/nen-en-iec-61851-1-2019-en-261254>.
26. Cai, H., Chen, Q., Guan, Z. & Huang, J. *Day-ahead optimal charging/discharging scheduling for electric vehicles in microgrids*. *Protection and Control of Modern Power Systems* **3**. ISSN: 23670983 (2018).
27. Liu, Z. et al. *Optimal Day-Ahead Charging Scheduling of Electric Vehicles Through an Aggregative Game Model*. *IEEE Transactions on Smart Grid* **9**, 5173–5184. ISSN: 19493053 (2018).
28. Baringo, L., Carrión, M. & Domínguez, R. *Electric Vehicles and Renewable Generation - Power System Operation and Planning Under Uncertainty* 578. ISBN: 9783031090783 (2023).
29. Yu, Y., Shekhar, A., Chandra Mouli, G. R. & Bauer, P. Comparative Impact of Three Practical Electric Vehicle Charging Scheduling Schemes on Low Voltage Distribution Grids. *Energies* **15**. ISSN: 19961073 (2022).
30. Haider, S., Rizvi, R. e., Walewski, J. & Schegner, P. Investigating peer-to-peer power transactions for reducing EV induced network congestion. *Energy* **254**, 124317. ISSN: 03605442. <https://doi.org/10.1016/j.energy.2022.124317> (2022).
31. Koltsaklis, N. E. & Knápek, J. Assessing flexibility options in electricity market clearing. *Renewable and Sustainable Energy Reviews* **173**. ISSN: 18790690 (2023).
32. Bañol Arias, N., Hashemi, S., Andersen, P. B., Træholt, C. & Romero, R. Assessment of economic benefits for EV owners participating in the primary frequency regulation markets. *International Journal of Electrical Power and Energy Systems* **120**, 105985. ISSN: 01420615. <https://doi.org/10.1016/j.ijepes.2020.105985> (2020).

33. Einolander, J. & Lahdelma, R. Explicit demand response potential in electric vehicle charging networks: Event-based simulation based on the multivariate copula procedure. *Energy* **256**, 124656. ISSN: 03605442. <https://doi.org/10.1016/j.energy.2022.124656> (2022).
34. GOPACS. *GOPACS - the platform to solve congestion in the electricity grid*. 2023. <https://en.gopacs.eu/>.
35. European Parliament and European Council. *Directive 2014/35/EU of the European Parliament and of the Council of 26 February 2014 on the harmonisation of the laws of the Member States relating to the making available on the market of electrical equipment designed for use within certain voltage limits* 2014.
36. International Organization for Standardization (ISO). *ISO 15118-20:2022 Road vehicles — Vehicle to grid communication interface — Part 20: 2nd generation network layer and application layer requirements*. <https://www.iso.org/standard/77845.html> (2022).
37. European Parliament and European Council. *Regulation (EU) 2018/858 of the European Parliament and of the Council of 30 May 2018 on the approval and market surveillance of motor vehicles and their trailers, and of systems, components and separate technical units intended for such vehicles, amending Regulations (EC) No 715/2007 and (EC) No 595/2009 and repealing Directive 2007/46/EC* 2018.
38. Python Software Foundation. *Python 3.9.12* 2022. <https://docs.python.org/3/reference/>.
39. Gurobi. *Gurobi Optimizer 9.5.2* 2022. <https://www.gurobi.com/>.
40. Delft High Performance Computing Centre (DHPC). *DelftBlue Supercomputer (Phase 1)* <https://www.tudelft.nl/dhpc/ark:/44463/DelftBluePhase1>. 2022.
41. Utrecht University - Faculty of Geosciences. *HPC cluster Eejit* <https://eejit-doc.geo.uu.nl>. 2023.
42. ENTSO-E. *ENTSO-E Transparency Platform* 2023. <https://transparency.entsoe.eu/>.

Reviewer #1	Response
	Dear reviewer, Thank you for your constructive review comments. We believe that due to your comments, the quality of the manuscript has improved considerably.
-justify the selection of the currents given in Fig. 1. why these current ranges are selected.	Figure 1 presents the considered charging currents for the different conducted smart charging tests. These smart charging tests aim to test whether the EV and EV charging stations can handle four different situations which could occur with different applications of EV smart charging. The smart charging tests were conducted at the internationally-renowned Testlab of ElaadNL, following a standardized testing protocol. The specific parameters of these smart charging tests (e.g., selected current, duration of pauses etc.) have been determined by ElaadNL in close consultation with different industrial stakeholders involved in actual EV smart charging pilots in the Netherlands (i.e., grid operators, car manufacturers, EV charging station manufacturers). As Figure 1 shows, charging currents of either 6 or 32 amperes were regularly considered in the smart charging tests. These currents refer to the minimum and maximum charging current for AC EV charging, as defined in the International Electrotechnical Commission's communication standard for EV charging (IEC 61851). Inspired by this comment, we have thoroughly revised the text in the ‘Results – Technical smart charging tests’ section and included more details about the technical smart charging tests that were conducted and their relevance. The paragraph discussing these aspects now reads as follows: The smart charging tests conducted at the Testlab aim to evaluate the EV's response when exposed to various charging patterns that could occur with different applications of smart charging. The parameters of the charging tests have been determined in consultation with different stakeholders, such as grid operators and EV manufacturers. The fluctuating charging test assesses whether an EV can handle smart charging applications with high fluctuations in the charging signal, such as solar charging (i.e., directly linking the charging power to the solar generation of photovoltaic (PV) systems). As shown in Figure 1, this test considers a fluctuating charging signal between 6 and 32 amperes at a 60-second interval. These current values correspond to the prescribed minimum and maximum charging currents for EV charging, as defined in the International Electrotechnical Commission's communication standard for EV charging [23]. The intermittent charging test evaluates the EV's response to a charging session with a high number of charging pauses, which is for instance relevant when applying smart charging for load balancing at car parks (i.e., quickly alternating the charging power between charging stations to reduce the peak charging power of the car park). In this test, the charging signal is switched 25 times between 0 and 6 amperes at 60-second intervals. The last set of tests analyses the EV's response to paused and delayed charging. These tests are relevant for smart charging applications that require longer periods without charging, such as smart charging to reduce charging costs with static or dynamic ToU tariffs or smart charging to mitigate grid congestion. The paused and delayed charging tests have a similar setup. The paused charging tests assess the EV's ability to properly react to the charging signal after a charging pause, which is implemented after the vehicle has been charged for a brief period. The delayed charging tests also consider a charging pause, which starts directly after the EV arrives at the charging station. Both tests are conducted with pauses of 20 minutes and 6 hours.

-Is the aim of this work only raising awareness? as it is written in the last paragraph of intro. What about solutions regarding these challenges? Some are already given. Hence, elaborate on the contribution more in this section.	Thank you for this suggestion. Inspired by this comment, we have elaborated on the contributions of this work in the introduction, highlighting that we quantify the impact of EVs' inability to handle charging pauses on the effectiveness of EV smart charging and that we present different solutions for these challenges (last sentence). The contributions paragraph now reads as follows: In this work, we shed light on these problems to raise awareness among relevant stakeholders (e.g., grid operators, EV manufacturers and policymakers) about the prevalence of technical smart charging problems and their impact on the effectiveness of smart charging. We present the results of large-scale technical charging tests, which indicate that around one-third of the EV models in the market cannot handle charging pauses or delays. Moreover, this study presents the results of model simulations that quantify the impact of EVs' inability to perform paused or delayed charging on three different applications for which smart charging can be used, namely: i) charging cost reduction, ii) mitigation of grid congestion, and iii) offering flexibility products to grid operators. The outcomes of these model simulations show that the potential impact of smart charging is halved for all applications if paused or delayed charging cannot be considered. Lastly, the current international regulations and standards on this topic are discussed, and options to eliminate paused and delayed charging problems are analysed.
-EV can handle 6 and 32 amperes. Is it for all type of EVs? What about the size of EVs? There is a need to conduct sensitivity analysis.	Most EV models and EV charging stations accept charging currents between 6 and 32 amperes. This minimum current is defined in the IEC 61851-1 communication standard between the EV and the charging station. It states that a minimum charging current of 6 amperes is required before an EV is allowed to start charging. This is why we considered the minimum charging current of 6 amperes in this work. The maximum charging current of 32 amperes is also based on this standard. This standard defines that the maximum charging current for all AC charging modes (i.e., non-fast charging modes) equals 32A. The on-board chargers of some EV models limit the charging power with higher charging currents. For instance, the Tesla Model 3 limits itself to 16A when using 3-phase AC charging (e.g., see this link). This did not affect our analysis. As described in the section named 'Methods – Model simulations – data inputs & preparation', this study considered historical EV charging session data and the maximum charging power of a charging session was obtained from the highest measured charging power in a session, and not based on the maximum allowed charging current of 32A. Hence, if a vehicle limits the charging power when exposed to a charging current of 32A, this is reflected in the observed maximum charging power in the used data. The minimum charging current of 6A was used to compute the minimum charging power of a vehicle (see the section named 'Methods – Model simulations – data inputs & preparation'). As discussed above, all vehicles should be able to charge with at least 6A according to the IEC 61851-1 charging standard, which means that the minimum charging power of an EV can be derived with high accuracy from the minimum charging current of 6A. For these reasons, we believe that there is no direct need to conduct an additional sensitivity analysis. This reviewer's comment has motivated us to provide additional explanation about the charging limits for EV models in the manuscript. We have included the following short explanation in the section named 'Results – Technical smart charging tests' of the revised manuscript: As shown in Figure 1, this test considers a fluctuating charging signal between 6 and 32 amperes at a 60-second interval. These current values correspond to the prescribed minimum and maximum charging currents for EV charging, as defined in the International Electrotechnical Commission's communication standard for EV charging [23].
-When there is minimum charging current; what about charging time	This is an important point that is being addressed. Our research demonstrates that incorporating charging pauses significantly enhances the efficiency of various smart charging applications. However, the introduction of charging pause may potentially trigger range anxiety concerns

optimization? Charging speed is critical for some customers, not having time. Fast charging etc. Hence, discussion on the charging times and its effect on the customer satisfaction should be provided.	among EV users. In scenarios where a minimum charging current of 6A is consistently considered after connecting an EV to the charging station, and an EV owner departs from the station prior to the estimated departure time, it is ensured that the EV has at least partly been charged, as the vehicle was continuously supplied with a charging current of at least 6A. On the other hand, if charging pauses are implemented and the connection duration of an EV to the charging station is not accurately predicted, there is a risk the EV has hardly been charged at departure, resulting in reduced customer satisfaction. To further address this point, we have renamed the ‘Conclusions’ section into ‘Discussion’. This section has been fully revised and different discussion points have been added to this section, including the discussion point brought up in this reviewer comment: Although this research indicated that it is important that EVs are able to deal with paused and delayed charging, it should be acknowledged that actual implementation of paused and delayed charging could trigger range anxiety issues among EV users. When scheduling the charging of an EV, its departure time from the charging station has to be estimated through user input and/or by applying forecasting methods. If an EV departs from the charging station before the anticipated departure time and the vehicle has continuously been charged with a charging current of at least 6 amperes, it is ensured that the EV has at least partly been charged. However, with the application of paused and delayed charging, there is a risk that the EV will receive a minimal charge if it departs before the expected departure time. Therefore, smart charging operators must exercise caution regarding the uncertainties in their models when employing paused and delayed charging scheduling. The reader should bear in mind that this additional uncertainty was not considered in this work’s model simulations for paused and delayed charging. Nevertheless, it should be realized that this real-world challenge may be largely mitigated by actively requesting user information about their charging sessions (e.g., expected departure time & charging demand), for instance through a mobile application (e.g., [37,38]), either as user-defined defaults with opt-outs or as per-session preferences.
-Based on the conducted simulation, i suggest to add few bullet points for policy making in charging.	The section named ‘Results - Options for eliminating charging problems’ discusses a range of enforcement or stimulation options that governments and policymakers could use to eliminate the delayed and paused charging problems of EVs. These enforcement and stimulation options were not presented as bullet points, as we discuss each option in a separate paragraph, and the text following the bullets would become very long. We believe that this would affect the flow and readability of the manuscript, which is why we decided to not use bullet points when discussing these options. We would also like to highlight that the discussion of policy options has been considerably extended compared to the previous version of the manuscript. This discussion now reads as follows: If the public sector, including governments and regulatory agencies, deem that the paused and delayed charging problems are too severe to be resolved without intervention, there are multiple enforcement or stimulation methods available. First, these organizations could stimulate manufacturers to comply with the existing standards that address this issue. This can be done by taking an active role in informing manufacturers about the importance of EVs being able to handle paused and delayed EV charging. Alternatively, the public sector has the option to establish an EV model certification program, where EV models that successfully completed a set of EV smart charging tests are granted a certificate, which could make the specific EV model more appealing to consumers.

	The public sector can also enforce the elimination of paused and delayed charging problems by implementing regulations on this topic. This could be modelled after the type-approval system that is currently in place. Before getting access to public roads in the EU, all car models need to acquire type-approval [36], issued by a national approval authority. The type-approval tests assess whether the car models meet EU safety rules (e.g., crash tests) and noise and emission limits. Expanding these tests to include an evaluation of the technical charging capabilities of EVs would ensure that only EV models meeting technical charging standards are permitted on the road. If the public sector prefers not to incorporate technical charging tests into the type-approval process, they could introduce legislation that makes compliance with standards that address the paused and delayed charging problems, such as ISO 15118-20, compulsory, either through self-assessment or by requiring testing and approval by a national approval authority. Ideally, all discussed enforcement or stimulation methods should be implemented at an international level, within entities like the EU, to enhance efficiency and maintain consistency in policies across different nations.
-Is it possible to validate the model in real environment?	The section named ‘Methods – Model simulations – charging models’ presents charging models in which a charging session cannot be paused and EVs should continuously be charged with the required minimum charging current of 6A. Such models have already been validated in practice: different real-world smart charging experiments do not consider charging pauses to avoid the charging problems discussed in this work. Hence, these real-world experiments considered a similar model formulation as the model formulation in this work, validating the models in a real-world environment. This is for instance described in the following paper: Brinkel, N., Markotić, P., Kuiper, L., Warmerdam, S., Baeten, B., Meersmans, J., ... & AlSkaif, T. (2023, June). Dynamic Grid Tariffs for Electric Vehicle Charging: Results from a Real-World Experiment. In 2023 IEEE Belgrade PowerTech (pp. 1-6). IEEE. This work describes the results of a real-world smart charging experiment. As described in this work, no charging pauses could be considered when scheduling the EVs, and a similar model formulation was adopted when scheduling EV charging in this experiment. This has been further highlighted in the manuscript, in the section named ‘Methods – Model simulations – charging models’: The validity of this model has been confirmed through real-world application [42] and the model is formulated as follows:
-Add discussion on renewable based EV charging since the timings will be dependent on availability of solar, wind etc.	We agree that this requires further discussion. As highlighted in Figure 2, many EV models cannot deal with smart charging applications that consider fluctuating or intermittent charging profiles. This is problematic when applying solar charging, in which the charging power of an EV is directly linked to the output of a solar PV system. However, most of the paper is focused on the impact of EVs not being able to perform delayed or paused charging. For this reason, we have added the following text to the renamed ‘Discussion’ section: In addition, it should be recognized that the model simulations in this research exclusively focused on quantifying the impact of the EV's inability to perform paused and delayed charging. The results of the technical smart charging tests indicated that fluctuating or intermittent charging problems also occur frequently. This could harm the roll-out of different smart charging applications, including solar charging, in which the EV charging power depends on the output of a PV system. Solar charging could contribute to increasing PV self-consumption and facilitate the grid integration of PV technology [40,41]. For this reason, policy addressing technical charging problems for EV smart charging should also encompass the resolution of technical charging problems related to fluctuating or intermittent charging signals.

Reviewer #2	Response
Please see the attached comments / suggested edits / queries on the PDF.	Dear reviewer, Thank you for your constructive review comments and thank you for providing a PDF with detailed comments. We believe that due to your comments, the quality of the manuscript has improved considerably.
I think the Abstract is fairly clear - but not fully so. Whose awareness should be raised, and with hat 'theory f change' for impact?	We have extended the abstract and further specified whose awareness should be changed, and what is the desired impact of this work. This is formulated as follows in the revised abstract: This research examines this problem to inform various stakeholders, including policymakers and manufacturers, and stimulates the adoption of proactive measures that address this problem.
In the Introduction: Really needs to say more about peaks for which type of EV charging context ['on-street', home, work-place'].	In this work, we mostly focus on charging in residential low-voltage grids, which happens at 'home' or 'on-street' charging stations. Inspired by this comment, we have further specified this in the introduction: Most EV charging occurs in Low-Voltage (LV) grids at home or on-street charging stations [3], and the majority of these grids were designed decades ago without the concept of EV charging in mind. The charging power of an EV is significantly higher than the typical peak-time power consumption of a household, and since most EV users tend to arrive at their charging station at a similar time, concentrated charging moments are expected in residential LV grids [4,5]. This has also been further outlined in the 'Methods' section.
Is it just EV owners, or (also) users - and how do you see the distinction between the two in your article / argument?	Thank you for addressing the inconsistencies in terminology here. In the original version of the manuscript, both references to "EV owners" and "EV users refer to the person driving the EV and charging it at the charging station. To improve consistency, we have replaced 'EV owners' with 'EV users' throughout the manuscript.
You really need to give the reader a definition of 'smart charging'.	The introduction has been extended by adding the following definition of 'smart charging': This provides ample opportunities for EV smart charging. With smart charging, EV charging sessions are optimized for different objectives by aligning the charging moments and charging speed over time with user preferences and current market or grid conditions [3,13].
How do you mean by having paid little attention? Researchers? Regulators? Governments? Or perhaps [less likely, and if so more deliberate?] EV manufacturers?	Thank you for pointing out that our initial statement ' Remarkably, the technical problems associated with smart charging have received little attention ' could lead to confusion. The intention behind this statement is to convey that technical charging problems are not widely discussed in both scientific literature and the media, resulting in limited awareness among policymakers and EV manufacturers. To avoid confusion, this statement has been rephrased as follows:

	Remarkably, the technical problems associated with EV smart charging have hardly been addressed in scientific literature and the media, leading to low awareness about these issues among different stakeholders, including policymakers, EV manufacturers and grid operators.
Current legislation? Do you mean standards / regulation / legislation? At global/international level [IEC], EU?, national, regional/local?	In our initial statement in the introduction, we used the term ‘current legislation’ to refer to all regulations and standards that are currently in place on this topic. As highlighted in the section named ‘Results – options for eliminating charging problems’, the current regulations and standards on this topic have an international scope. Consequently, we have substituted the term ‘current legislation’ in the introduction with ‘current international regulations and standards’.
Results: see Figure 2: Is there a pattern of EV models to which you concerns apply: Hybrid EVs? Full EVs? Depending on EV battery capacity?	No clear pattern is observed regarding the specific types of EV models experiencing technical charging issues; these problems manifested with comparable frequency in both hybrid and battery electric vehicles. In addition, the occurrence of the technical charging problems seem to be independent of the battery capacity of the EV models; the technical charging problems occurred with similar frequency between EV models with a small and large battery capacity. It has been agreed with the EV manufacturers when performing the charging tests that the results could only be presented in an aggregated manner, to ensure that no individual EV model can be identified from the analysis. For this reason, we are not allowed to present more detailed results of the technical charging tests.
2.2.1. You should have mentioned mentioned [both] Time-of-Use and Dynamic Electricity [EV charging] Prices much earlier before - even in the Abstract.	We agree that our references to ‘dynamic electricity prices’ in the initial version were incomplete. This has been replaced throughout the manuscript by ‘static and dynamic time-of-use tariffs’. We believe that the term ‘static and dynamic time-of-use tariffs’ covers all applications for which EV smart charging can be used to reduce costs:  - Static time-of-use tariffs: Prices vary throughout the day, but in a repeating manner, e.g. fixed night and day tariffs. - Dynamic time-of-use tariffs: Prices vary throughout the day, but in a non-repetitive manner, e.g. day-ahead market prices Next to replacing ‘dynamic electricity prices’ by ‘static and dynamic time-of-use tariffs’, we have introduced ‘static and dynamic time-of-use tariffs’ earlier in the paper, in the introduction. This is the updated text in the introduction: Additionally, it can help EV users reduce their charging costs by taking advantage of moments with low electricity market prices when considering static or dynamic Time-of-Use (ToU) pricing schemes [19,20]. Given the strict word limit for the abstract of 150 words, we have not introduced static and dynamic time-of-use tariffs in the abstract.
2.2.2. I don't follow with the logical connections that "grid congestion solutions will fail with lower EV adoption rates". Why / how so?	The initial version of the manuscript contained the following statement: “When a minimum charging current needs to be applied, EV charging cannot be completely shifted away from peak hours. Hence, grid congestion solutions will fail with lower EV adoption rates.”. We acknowledge the reviewer's concern about the clarity of the final sentence in this statement. Our intent in that sentence was to convey that the application of smart charging to mitigate grid congestion problems which are induced by EV charging is less effective if no paused or delayed charging can be applied, as the EV charging cannot be fully shifted away from peak hours. As a result, grid congestion

	problems are more likely to manifest at lower EV adoption levels when smart charging does not involve paused or delayed charging, in contrast to a scenario where such features are incorporated. To improve the clarity of the manuscript, this statement has been rewritten as follows: Smart charging can also be used to address grid congestion problems induced by EV charging by shifting the charging from moments with high local grid load to moments with low local grid load [28,29]. When charging pauses cannot be considered, EV charging cannot be completely shifted away from peak hours. Consequently, grid congestion problems will manifest at lower EV adoption levels when deploying smart charging without paused or delayed charging compared to the deployment of smart charging with these features.
Figure 4: Which LV grid A Dutch one? [where?]	The analysis used to create Figure 4 considered an LV grid in a residential area in the city of Utrecht, the Netherlands. This has been further specified in the caption of the figure in the revised manuscript.
2.2.3. Look at the grammar in the first sentence.	The first sentence of this section. has been reformulated as follows: “Moreover, smart charging can be applied to offer flexibility services to grid operators, for instance by supplying balancing reserves (e.g., frequency restoration reserves) to TSOs for restoring the balance between supply and demand [30-32].”
2.3 Needs a bit re-writing in more precise language - see my queries on the PDF.	Thank you for providing the very helpful and detailed comments on the PDF. We have thoroughly checked your comments and have implemented changes in all sections. The changes are highlighted in the ‘revised manuscript with changes highlighted’ document.
Figure 5: which types of charging stations, and where?	This analysis used data from on-street charging stations located in the city of Utrecht, the Netherlands. This has been further specified in the caption of this figure.
Will most EV models "soon" be able to perform V2G / bi-directional charging?	This is an interesting point. There are different aspects that will affect the future roll-out of V2G/bi-directional charging:  1. The first production-scale EV models that support AC V2G were recently introduced to the market (e.g., Hyundai IONIQ5), but the majority of car models that are currently introduced to the market are not V2G-compliant. 2. Not all AC charging stations are V2G compliant, although this is now a requirement in the tenders for public charging stations in different European cities, including Utrecht, the Netherlands. 3. Only a small share of the car models are currently compliant with the ISO15118-20 standard that is required to perform bi-directional charging. 4. There are some regulatory issues (e.g., certification of EVs by grid operators before the EV can supply electricity to the grid) which need to be tackled before large-scale implementation of V2G. This shows that a separate study could be dedicated to the barriers to and current status of the roll-out of V2G technology. As we aim to keep this work concise, we have decided to not include this discussion into this work.

Think a bit more about at which level [EU / national] these potential voluntary agreements with manufacturers?	Thank you for addressing this interesting point. We believe that all measures described in the section named ‘Results – options for eliminating charging problems’ are ideally taken on an international level (i.e., EU level), to enhance efficiency and avoid discrepancies in the policies between different nations. This has been further specified in the manuscript: Ideally, all discussed enforcement or stimulation methods should be implemented at an international level, within entities like the EU, to enhance efficiency and maintain consistency in policies across different nations.
Now you are talking about EV "buyers", previously "owners and at times "users". Work this through and clarify this right through. And a significantly share of EVs is now leased ...	Thank you. We have increased the consistency in terminology in this work by replacing the terms ‘EV buyers’ and ‘EV owners’ by ‘EV users’ throughout the manuscript.
Be clearer about where EV charging costs are constituted / come about. Conceptually, perhaps this might stimulate thinking: https://www.sciencedirect.com/science/article/pii/S0306261919316526	Thank you for addressing this point. We have expanded the introduction to cost-optimized charging in the section named ‘Results - Impact on smart charging’s charging cost reduction potential’, in order to provide the reader with better understanding about the mechanisms behind this. The description is now as follows: The inability of EVs to perform paused or delayed charging can diminish the effectiveness of different smart charging applications, including smart charging for participating in an electricity market that considers static or dynamic ToU tariffs (e.g., day-ahead electricity market) [24-26]. Charging costs can be reduced by shifting the charging demand from moments with high prices to moments with low prices. The provided paper was relevant and a reference to this work has been included in the manuscript.

Reviewer #3	Response
This paper investigates the minimum charging power issue that may lock the potential of smart charging in reducing cost, mitigating grid congestion, and offering flexibility services. To prevent EVs from shifting to sleep mode when charging is interrupted, a minimum charging power is necessary, even at moments when a charging pause is desirable. This paper uses extensive model simulations to show the impact of this technical issue on the benefits of smart charging in the above scenarios. A quantitative analysis is conducted from the real-world data, showing that almost half of the potential benefits are	Dear reviewer, Thank you for your constructive review comments. We believe that due to your comments, the quality of the manuscript has improved considerably.

lost. This paper aims to raise awareness of this technical issue and also proposes several legal and practical solutions.	
In general, this paper focuses on a small problem yet shows that its impact could be surprisingly huge. The three main plots from running different charging models on the real-world data are informative and offer nice insights. However, I have major concerns about the technical contributions of this work. First, the minimum charging power is easy to model and incorporate into most of the EV charging literature. Therefore, technically it is not a novel problem.	We agree that this paper focuses on a very specific problem, but as the results of this work report a tremendous impact of this problem on the effectiveness of EV smart charging applications, we believe that it is important to dedicate a focused study on it. The authors agree that introducing a minimum charging power constraint to an optimization model to avoid charging pauses will generally be regarded as a relatively minor challenge: One needs to include binary variables and a minimum charging power constraint to the optimization problem (see the section named ‘Methods - Model simulations - charging models’) to assure that i) EVs charge continuously with at least the minimum charging power after their arrival to the charging station until their charging demand is satisfied, and ii) the minimum charging power constraint no longer applies when the charging demand of a charging session is satisfied. This is exactly the main point of this work: even though it is not extremely challenging to incorporate the continuously-required minimum charging power into charging optimization models, it is neglected in all scientific literature on the topic of EV smart charging. In this study, we aim to show, through extensive model simulations, that most literature in this field overestimates the potential of smart charging and that more realistic insight into the current potential of smart charging can be obtained by making a relatively simple alteration to charging models (i.e., introducing a continuously-required minimum charging current to prevent charging pauses). This in turn demonstrates the impact of these constraints, resulting in a call for action (in the section named ‘Results - Options for eliminating charging problems’ and the updated discussion section).
Second, the three charging models are very standard offline formulations to schedule EV charging. However, they are not directly implementable in an online fashion in practice. Thus, the computed reduction in potential benefits is, at most, a lower bound. It may not reflect the actual impact of the minimum charging power.	Thank you for this remark. We indeed considered an offline (i.e., not real-time) model formulation for scheduling EV charging in this work. Delayed and paused charging problems apply to both offline and online charging. Even when scheduling EV charging real-time (i.e., online formulation), charging pauses still cannot be considered, to avoid that EVs switch to sleep mode and become unresponsive to charging signals. Hence, this means that EVs with an online model formulation still need to charge continuously with at least the minimum charging current at moments at which they ideally would not charge. The main difference between an online and offline model formulation will be the uncertainty associated with the charging session characteristics (e.g., departure time and charging demand) of each session. These charging session characteristics are generally assumed to be known in this study prior the optimisation with an offline model formulation. This was also the case in this study. When optimising a charging session in practice, this often happens real-time (i.e., online formulation) and there will be more uncertainty regarding these charging session characteristics. To avoid that EV users face an empty battery if they (unexpectedly) depart from the charging station before the anticipated departure time, the applied charging schedules with real-time charging scheduling might be more conservative to reduce the risk of user dissatisfaction. However, for two reasons, we believe that the impact of using an offline model formulation to evaluate the consequences of EVs' inability to perform paused or delayed charging is relatively minor. Firstly, in many cases, departure information may be

	known at the time a car connects through explicit user information (e.g. mobile applications in which users communicate their expected departure time and charging demand). In these cases, the difference in uncertainty between the offline and online model formulation is considerably reduced, and EVs will not necessarily be charged more conservatively in the online formulation. Secondly, the increased uncertainty associated with an online model formulation also holds true in scenarios where paused and delayed charging cannot be applied. Hence, EVs will be charged more conservatively with the online model formulation compared to the offline model formulation, regardless of whether delayed and paused charging can or cannot be considered. To further address the uncertainty aspect of uncertainty in the manuscript, we have added a discussion to the discussion section: The reader should bear in mind that this additional uncertainty was not considered in this work's model simulations for paused and delayed charging. Nevertheless, it should be realized that this real-world challenge may be largely mitigated by actively requesting user information about their charging sessions (e.g., expected departure time & charging demand), for instance through a mobile application (e.g., [37,38]), either as user-defined defaults with opt-outs or as per-session preferences.
Last, the discussion of the options for eliminating this charging problem is a bit abstract. The enforcement methods from governments/legislation, charging operators, and consumers are briefly mentioned, but details are missing.	Inspired by this comment, we have considerably expanded the discussion of the different enforcement methods that governments could use to eliminate paused and delayed charging problems. This is highlighted in the ‘revised manuscript with changes highlighted’. We have added a concrete example of a policy initiative to stimulate the elimination of paused and delayed charging problems, by explaining how governments could introduce a certification scheme for this. In addition, we have further explained how the smart charging performance of EV models could be linked to the type approval system. Moreover, we have included a discussion about the preferred level (national/international) of implementation of the different proposed policies. This section now reads as follows: If the public sector, including governments and regulatory agencies, deem that the paused and delayed charging problems are too severe to be resolved without intervention, there are multiple enforcement or stimulation methods available. First, these organizations could stimulate manufacturers to comply with the existing standards that address this issue. This can be done by taking an active role in informing manufacturers about the importance of EVs being able to handle paused and delayed EV charging. Alternatively, the public sector has the option to establish an EV model certification program, where EV models that successfully completed a set of EV smart charging tests are granted a certificate, which could make the specific EV model more appealing to consumers. The public sector can also enforce the elimination of paused and delayed charging problems by implementing regulations on this topic. This could be modelled after the type-approval system that is currently in place. Before getting access to public roads in the EU, all car models need to acquire type-approval [36], issued by a national approval authority. The type-approval tests assess whether the car models meet EU safety rules (e.g., crash tests) and noise and emission limits. Expanding these tests to include an evaluation of the technical charging capabilities of EVs would ensure that only EV models meeting technical charging standards are permitted on the road. If the public sector prefers not to incorporate technical charging

	tests into the type-approval process, they could introduce legislation that makes compliance with standards that address the paused and delayed charging problems, such as ISO 15118-20, compulsory, either through self-assessment or by requiring testing and approval by a national approval authority. Ideally, all discussed enforcement or stimulation methods should be implemented at an international level, within entities like the EU, to enhance efficiency and maintain consistency in policies across different nations.
I also have a few other comments below. 1. The authors highlighted delayed charging problems in the paper. However, given Figure 1, it is a bit confusing since the "delay charging" test seems the least relevant (no pause or intermittency). I suggest using a more clear terminology.	Thank you for raising this point. The delayed charging test is very similar to the paused charging test. In both tests, a charging pause is considered. The difference lies in the moment of the charging pause. With the paused charging test, the charging is paused after a vehicle has briefly charged. With the delayed charging test, the charging is paused directly after the arrival of the EV to the charging station. This could be relevant for cases in which the EV arrives at the charging station at moments with high electricity prices or a high grid load. The difference between both tests has been further specified in the manuscript: The last set of tests analyses the EV's response to paused and delayed charging. These tests are relevant for smart charging applications that require longer periods without charging, such as smart charging to reduce charging costs with static or dynamic ToU tariffs or smart charging to mitigate grid congestion. The paused and delayed charging tests have a similar setup. The paused charging tests assess the EV's ability to properly react to the charging signal after a charging pause, which is implemented after the vehicle has been charged for a brief period. The delayed charging tests also consider a charging pause, which starts directly after the EV arrives at the charging station. Both tests are conducted with pauses of 20 minutes and 6 hours. In addition, we would like to point out that, as mentioned in the opening of this response letter, we have replaced 'delayed charging' by 'paused and delayed charging' throughout the manuscript.
2. From the smart charging test results (Figure 2), what is the percentage of unsuccessful tests that were caused by charging currents exceeding charging signals? This unsuccessful label does not seem relevant to the necessity of minimum charging power.	The reason why the testing procedure for the smart charging tests identifies a charging test as unsuccessful if the charging current of the EV exceeds the charging signal by 0.5 amperes, is that this is a violation of the EV communication standards (IEC61851-1). These state that the charging current may never exceed the charging signal. If the charging station recognises this, it could terminate the charging session. This is because if this current overshoot is too severe or occurs too frequently, main circuit breakers might break down because of these current violations. This problem only occurred with the intermittent and fluctuating charging tests, and did not occur with the paused and delayed charging tests. For the intermittent and fluctuating charging tests, 80% of the charging tests that failed were caused by current violations. Inspired by this comment, we have made two alterations to the manuscript. First, we have further explained in the section named 'Results - Technical smart charging tests'. why a charging test failed if the charging current was violated by at least 0.5 amperes, and that this only applies to the fluctuating and intermittent charging tests:

	A charging test was labelled as unsuccessful if the tested EV model ceased charging or if its charging current violated the current limits specified in the EV charging standards by exceeding the charging signal by at least 0.5 amperes [23]. The latter problem only occurred with the fluctuating and intermittent charging tests. Secondly, we specified in the same section that most failed tests for the intermittent and fluctuating charging tests were caused by current limit violations. The success rate of the fluctuating charging test equals 71% for the tested EV models. The share of tested models that can follow the intermittent charging profile is lower and equals 63%. In both cases, most failed tests were caused by current limit violations.
3. The three charging models are also stylized. If the authors consider discrete charging power instead of continuous one, I suspect the results would be much different.	This study indeed assumed a continuous charging power; the charging power in the considered optimization model can take any value between the minimum and maximum charging power of a charging session (see the section named ‘Methods - Model simulations - charging models’). We acknowledge that a few scientific studies have presented discrete EV charging models (e.g., see link and link). However, all these studies justified their proposition that EVs can only be charged discretely based on one specific reference that dates back to 2012: Gan, L., Topcu, U., & Low, S. H. (2012, July). Stochastic distributed protocol for electric vehicle charging with discrete charging rate. In 2012 IEEE Power and Energy Society General Meeting (pp. 1-8). IEEE. EV chargers have significantly advanced since 2012 and currently all chargers in the market can handle a continuous charging power. This has also been confirmed by the technical experts in the ElaadNL Testlab. As outlined in the section named ‘Methods - Model simulations - charging models’, these experts performed technical charging tests with a large share of the EV models and EV charging stations available in the market. This is also verified by consulting the current communication protocols for smart charging. The Open Charge Point Interface (OCPI) protocol is currently the standard communication protocol for smart charging. In this protocol, different parameters are exchanged between the smart charging operator and the charging station/EV. In the current version of the protocol (v2.2.1), only a minimum and maximum charging power or charging current need to be communicated. The smart charging operator can choose any value for the charging power/current between the provided minimum and maximum values. This means that when charging sessions are optimized in practice, continuous charging power can be considered in the optimization models. Although the models are all somewhat stylised, we believe that the continuous control of charging power reflects real-world charging behaviour in the considered context.

REVIEWER COMMENTS

Reviewer #1 (Remarks to the Author):

Most comments were addressed successfully.

2 additional points need further elaboration

1) Use of these smart charging and challenges for PHEV vehicles as well. How does it affect them?

2) The discussion on renewable-based EV smart charging is still not deep enough. Fluctuating nature and energy storage aspects should be also discussed.

Reviewer #2 (Remarks to the Author):

I find the Response Letter to the Reviewers / Editor both clear and well motivated and substantiated.

I also agree with all the changes made, and likewise with what has been retained.

Accordingly, I can now endorse this manuscript for publication.

(Regarding Reviewer #3's remaining concerns:)

Having read the latest version, I think the revised manuscript deals satisfactorily with the review requests by Reviewer #3, and I think the authors' point-by-point response can be accepted.

Dear Reviewers,

Thank you for your constructive review of the first revision of our submission of our manuscript “Unlocking the full potential of smart charging: Addressing paused and delayed charging problems in electric vehicles” to “Nature Communications”. We have addressed all comments provided by the reviewers while revising the manuscript. In the table below, we detail how we have dealt with the reviewers’ comments in the second revision of the manuscript.

Yours sincerely, on behalf of all authors,

Nico Brinkel

Reviewer #1	Response
Most comments were addressed successfully. 2 additional points need further elaboration	Dear reviewer, We highly appreciate your efforts in reviewing our manuscript once more. Your comments are valued, and we acknowledge that they enhanced the quality of our work. Below, we address the two points you raised in your review.
1) Use of these smart charging and challenges for PHEV vehicles as well. How does it affect them?	We acknowledge that the relationship between the type of EV (Plug-in Hybrid Electric Vehicles (PHEVs) and Battery Electric Vehicles (BEVs)) and the problems addressed in this work should be further highlighted in the manuscript. The results of the technical charging tests of this work showed that the charging problems outlined in this work manifested with comparable frequency in both PHEVs and BEVs. Different changes have been implemented in the manuscript to further highlight this:  1. First, it has been further highlighted in the Introduction: However, it is not widely known that the current deployment of smart charging is hindered by the technical capabilities of EVs. As this research will show, a significant share of EV models (both Plug-in Hybrid EV (PHEV) and Battery EVs (BEV) models) is unable to perform paused or delayed charging. In paused charging, the charging process is interrupted after the EV was previously charging, while in delayed charging, the start of the charging process is postponed after the EV arrives at the charging station. 2. Second, it has been further emphasized in the section titled ‘Results – Technical smart charging tests’ that both PHEVs and BEVs were tested in the technical smart charging tests: The charging tests have been performed on a large share of the PHEV and BEV models on the Dutch market and manufacturers can use the results of these tests to improve the technical performance of their products. 3. It has been highlighted twice in the same section that the charging problems occurred both at PHEVs and BEVs:

	The success rate of the fluctuating charging test equals 71% for the tested EV models. The share of tested models that can follow the intermittent charging profile is lower and equals 63%. In both cases, charging problems were observed in both PHEVs and BEVs, with the majority of failed tests attributed to violations of current limits. When considering a charging pause of 20 minutes, the success rate for the tested EV models equals 86% and 83% for paused and delayed charging, respectively. When the pause duration is extended to 6 hours, the share of tested EV models that successfully pass the charging test reduces to 71% and 67% for paused and delayed charging, respectively. These problems manifested with both PHEVs and BEVs. 4. Lastly, the methods on the technical smart charging tests have been slightly extended, by outlining that both PHEV and BEV models were tested: A large majority of the sold EV models (both PHEV and BEV models) in the Netherlands have undergone the technical charging test procedure at the Testlab. This study reported the results of the technical smart charging tests that were conducted at the Testlab between 1 June 2020 and 1 January 2023. Detailed charging test results cannot be disclosed individually due to non-disclosure agreements with EV manufacturers. The model simulations in this work were conducted using a dataset of charging sessions occurring at public, on-street charging stations in the city of Utrecht, the Netherlands (as outlined in the ‘Methods’ section). These charging stations are used by both PHEVs and BEVs and hence, both PHEV and BEV charging sessions were included in the data set. Since the EV model and the type of EV was not identified in the charging session data, it was not possible to make a distinction between PHEVs and BEVs in the model simulations. Inspired by this comment, the following adjustment has been made to the section titled ‘Model simulations — data inputs & preparation’: 1. All model simulations only considered charging session data from public, on-street charging stations located in the city of Utrecht, the Netherlands. These stations were accessible to both PHEVs and BEVs.
2) The discussion on renewable-based EV smart charging is still not deep enough. Fluctuating nature	Thank you for addressing this point. We have implemented different changes to the manuscript based on this comment:

and energy storage aspects should be also discussed.	 1. First, we have extended the introduction by introducing renewable-based EV smart charging when discussing the different applications for smart charging: Lastly, the roll-out of smart charging could accelerate the energy transition by shifting the charging demand of EVs to moments with excess renewable generation [21-23], thereby reducing the dependency on fossil-based energy resources and mitigating the intermittency challenges associated with renewable energy sources. If vehicle-to-grid (V2G) functions are considered, EVs could even act as a storage medium for excess renewable energy, which can be utilized to meet the electricity demand during periods of renewable energy shortage. 2. The Discussion section has been extended by further discussing how renewable-based EV smart charging could mitigate the problems associated with the intermittency of renewables: This could harm the roll-out of different smart charging applications, including renewable-based charging, in which the EV charging power depends on the output of a PV system or wind turbine. Implementing renewable-based charging systems has the potential to boost the self-consumption of renewable energy and enhance the integration of renewable energy technologies into the grid [21-23]. This approach helps mitigate the intermittency of renewable energy generation and reduces dependence on fossil fuels to fulfil electricity demand. For this reason, policy addressing technical charging problems for EV smart charging should also encompass the resolution of technical charging problems related to fluctuating or intermittent charging signals.
---	--

Reviewer #2	Response
I find the Response Letter to the Reviewers / Editor both clear and well motivated and substantiated. I also agree with all the changes made, and likewise with what has been retained. Accordingly, I can now endorse this manuscript for publication.	Dear reviewer, We highly appreciate your efforts in reviewing our manuscript once more. We are also grateful for your review of our response to the comments from reviewer 3. Your positive evaluation is greatly appreciated and we acknowledge that your comments from the first revision round enhanced the quality of our work.

(Regarding Reviewer #3's remaining concerns:)

Having read the latest version, I think the revised manuscript deals satisfactorily with the review requests by Reviewer #3, and I think the authors' point-by-point response can be accepted.

REVIEWERS' COMMENTS

Reviewer #1 (Remarks to the Author):

The authors have addressed the points in the second round successfully.

Dear Reviewer,

Thank you for your review of the second revision of our submission of our manuscript “Unlocking the full potential of smart charging: Addressing paused and delayed charging problems in electric vehicles” to “Nature Communications”. Your positive evaluation is greatly appreciated and we acknowledge that your comments from the first and second revision rounds enhanced the quality of our work.

Yours sincerely, on behalf of all authors,

Nico Brinkel

Reviewer #1	Response
The authors have addressed the points in the second round successfully.	Dear reviewer, We highly appreciate your efforts in reviewing our manuscript once more. Your positive evaluation is greatly appreciated and we acknowledge that your comments from the first and second revision rounds enhanced the quality of our work.